# Norrin/Frizzled4 signalling in the preneoplastic niche blocks medulloblastoma initiation

Erin A Bassett[1*†], Nicholas Tokarew[1,2†], Ema A Allemano[1], Chantal Mazerolle[1], Katy Morin[1], Alan J Mears[1], Brian McNeill[1], Randy Ringuette[1,3], Charles Campbell[1,3], Sheila Smiley[1], Neno T Pokrajac[4,5], Adrian M Dubuc[4,6], Vijay Ramaswamy[4,6,7,8], Paul A Northcott[4,6], Marc Remke[4,6], Philippe P Monnier[9,10], David Potter[11], Kim Paes[11], Laura L Kirkpatrick[11], Kenneth J Coker[11], Dennis S Rice[11], Carol Perez-Iratxeta[1,3], Michael D Taylor[4,6,7], Valerie A Wallace[1,2,5,10*]

[1]Regenerative Medicine Program, Ottawa Hospital Research Institute, Ottawa, Canada; [2]Department of Biochemistry, Microbiology and Immunology, University of Ottawa, Ottawa, Canada; [3]Department of Cellular and Molecular Medicine, University of Ottawa, Ottawa, Canada; [4]Department of Laboratory Medicine and Pathobiology, University of Toronto, Toronto, Canada; [5]Donald K. Johnson Eye Institute, Krembil Research Institute, University Health Network, Toronto, Canada; [6]Developmental and Stem Cell Biology Program, Arthur and Sonia Labatt Brain Tumor Research Centre, The Hospital for Sick Children, Toronto, Canada; [7]Division of Neurosurgery, Department of Surgery, University of Toronto, Toronto, Canada; [8]Division of Haematology/Oncology, The Hospital for Sick Children, Toronto, Canada; [9]Genetics and Development Division, Krembil Research Institute, University Health Network, Toronto, Canada; [10]Department of Ophthalmology and Vision Sciences, University of Toronto, Toronto, Canada; [11]Department of Ophthalmology, Lexicon Pharmaceuticals Inc., The Woodlands, United States

*For correspondence:
erinbassett@hotmail.com (EAB);
vwallace@uhnresearch.ca (VAW)

†These authors contributed equally to this work

Competing interests: The authors declare that no competing interests exist.

**Abstract** The tumor microenvironment is a critical modulator of carcinogenesis; however, in many tumor types, the influence of the stroma during preneoplastic stages is unknown. Here we explored the relationship between pre-tumor cells and their surrounding stroma in malignant progression of the cerebellar tumor medulloblastoma (MB). We show that activation of the vascular regulatory signalling axis mediated by Norrin (an atypical Wnt)/Frizzled4 (Fzd4) inhibits MB initiation in the $Ptch^{+/-}$ mouse model. Loss of Norrin/Fzd4-mediated signalling in endothelial cells, either genetically or by short-term blockade, increases the frequency of pre-tumor lesions and creates a tumor-permissive microenvironment at the earliest, preneoplastic stages of MB. This pro-tumor stroma, characterized by angiogenic remodelling, is associated with an accelerated transition from preneoplasia to malignancy. These data expose a stromal component that regulates the earliest stages of tumorigenesis in the cerebellum, and a novel role for the Norrin/Fzd4 axis as an endogenous anti-tumor signal in the preneoplastic niche.

## Introduction

The tumor microenvironment is comprised of many stromal cell types, including the vasculature, which is well known to promote the growth and propagation of established tumors

(*Calabrese et al., 2007*; *Hambardzumyan et al., 2008*; *Hanahan and Coussens, 2012*). It has also become increasingly clear that construction of a tumor-permissive stroma is a dynamic process influencing all stages of malignant progression, through the ongoing extracellular matrix (ECM) remodelling and recruitment or activation of angiogenic vascular cells, fibroblasts and immune cells (*Hanahan and Coussens, 2012*; *Hanahan and Weinberg, 2011*). While there is mounting evidence that a permissive niche is required during the earliest steps of tumor initiation, studying this early crosstalk requires adequate multistage models of carcinogenesis (*Barcellos-Hoff et al., 2013*). Medulloblastoma (MB), the most common malignant brain tumor in children, has become a paradigm for the study of pediatric tumors and primary brain tumors; however little is known about tumor/stromal communication in the early stages of MB. It is challenging to model these early events with human tumor biopsies or MB cell lines, as both represent late stage tumors and the latter frequently fail to maintain their in vivo characteristics (*Sasai et al., 2006*). An effective surrogate to test these interactions endogenously is the $Ptch^{+/-}$ mouse (*Goodrich et al., 1997*), a model of the human predisposition to MB (Gorlin syndrome) that, along with a subset of sporadic MB, belong to the Sonic hedgehog (Shh) subgroup (*Taylor et al., 2012*). $Ptch^{+/-}$ mice progress through well-defined stages of tumorigenesis in the cerebellum, beginning with the ectopic proliferation of granule neuron progenitor cells (GNPs), which form preneoplastic lesions on the cerebellar surface by two weeks of age. While most of these lesions regress, a minority undergo malignant transformation to MB (*Kessler et al., 2009*; *Oliver et al., 2005*). The environmental signals that co-operate with *Ptch* haploinsufficiency to regulate lesion induction and transformation are poorly understood.

Here, we modelled pre-tumor/stromal crosstalk during $Ptch^{+/-}$ MB evolution by manipulating the Norrin/Frizzled4 (Fzd4) pathway, an endogenous signalling axis that regulates vascular development in the cerebellum via neural/endothelial interactions (*Wang et al., 2012*; *Xu et al., 2004*; *Zhou et al., 2014*). We demonstrate that the preneoplastic niche is a potent modulator of $Ptch^{+/-}$ tumor initiation. Loss of vascular Norrin/Fzd4 signalling, either genetically or by short-term blockade, creates a tumor-permissive stroma that promotes the formation of preneoplastic lesions and their progression to malignancy. We show that activation of angiogenesis and stromal remodelling are key features of the oncogenic microenvironment that dramatically accelerates tumorigenesis in the $Ptch^{+/}$ cerebellum. This is the first study to describe a stromal component to the early stages of Shh-driven tumor initiation in the brain.

## Results

### *Ndp* is expressed in GNPs and mouse and human Shh-MB

To assess the stromal compartment at early stages of tumorigenesis in the $Ptch^{+/-}$ cerebellum, we sampled entire lesions at postnatal day (P) 14 by Collagen IV+ immunostaining, and observed an invasion of vasculature in a minority (24%; *Figure 1A*). Furthermore, lesion volume, which is a measure of neoplastic progression in the $Ptch^{+/-}$ model, was statistically larger in vascularized lesions (mean 0.18 mm³) compared to non-vascularized ones (mean 0.029 mm³, *Figure 1B*). These observations are notable, considering that only a minority of lesions undergo malignant transformation and continue to grow into tumors (*Kessler et al., 2009*; *Oliver et al., 2005*). To explore this pre-tumor/blood vessel interaction, we targeted Norrin signalling, a well-characterized regulator of neural-endothelial cell communication in the cerebellum (*Wang et al., 2012*; *Zhou et al., 2014*). Norrin (encoded by the X-linked gene *Ndp*) is a secreted atypical Wnt ligand that signals specifically through the Fzd4 receptor and the Lrp5/6 and Tspan12 co-receptors to activate the canonical Wnt pathway in endothelial cells (*Figure 1C*), where it is required for blood brain barrier (BBB) integrity in the cerebellum (*Junge et al., 2009*; *Wang et al., 2012*; *Xu et al., 2004*). Using male $Ndp^{-/y}$ mice carrying an *Ndp-lacZ* knockout (KO) allele (*Junge et al., 2009*), we examined the cerebellar expression profile of *Ndp*. Consistent with alkaline phosphatase-based reporter data (*Ye et al., 2010a*), we detected *Ndp* expression in the Purkinje cell layer, presumably in Bergmann glia (*Figure 1D*). We also detected *Ndp* expression in the cerebellar external granule layer (EGL), where GNPs, the Shh-MB cell of origin, reside throughout postnatal development (*Figure 1D*). Combined X-gal staining and immunohistochemistry (IHC) during the peak period of GNP proliferation at P7 revealed that *Ndp* expression is concentrated in the outer region of the Pax6+ EGL, in the proliferative phospho-histone H3 (PH3)+ compartment (*Figure 1D*). β-gal+ cells also overlapped with myelin basic protein

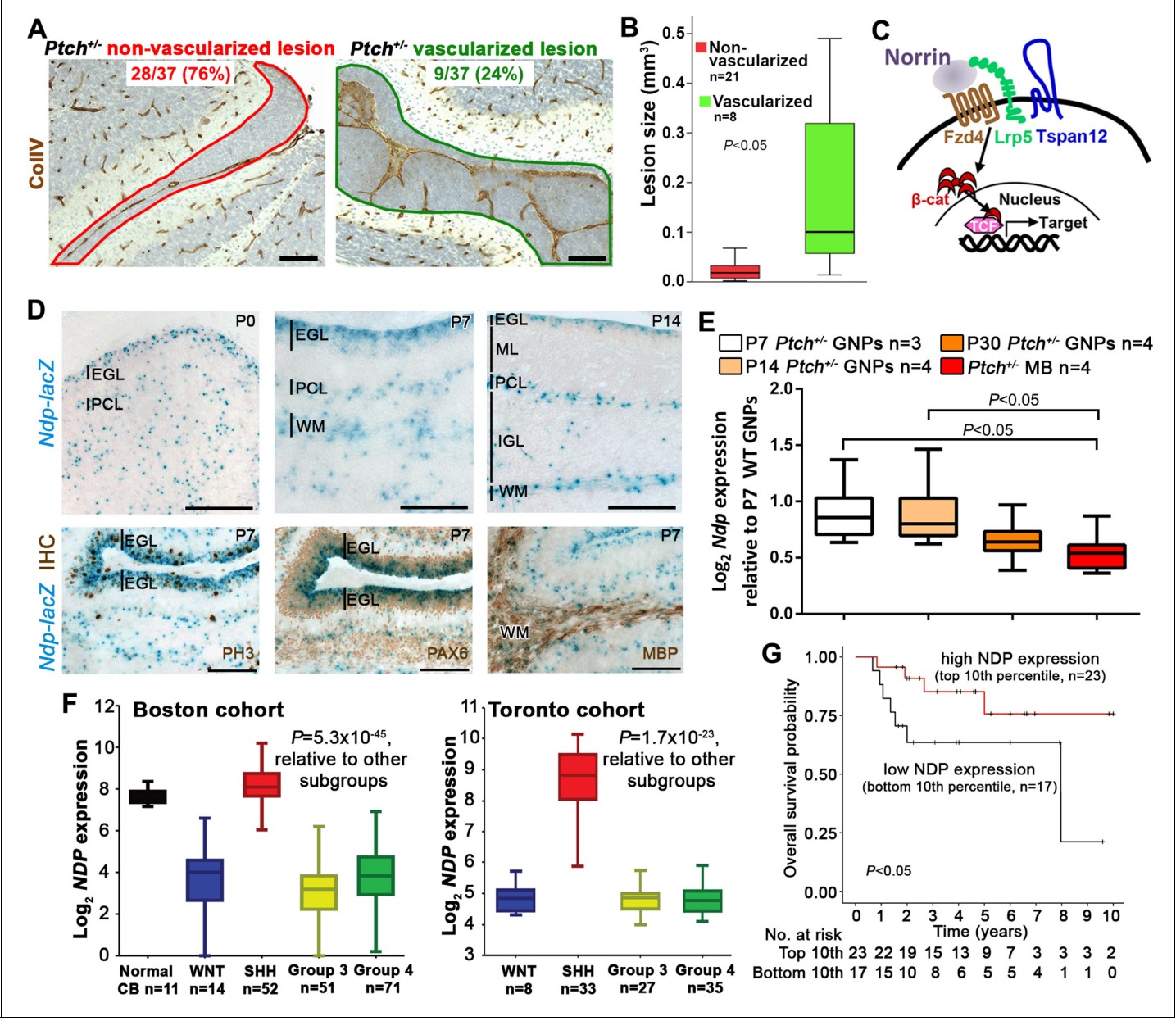

**Figure 1.** *Ndp* is expressed in Shh-MB precursors and mouse and human Shh-MB. (**A**) Representative immunostains for Collagen IV (ColIV) counterstained with hematoxylin, to depict non-vascularized (red outline) and vascularized (green outline) lesions from 37 *Ptch⁺/⁻* mouse cerebellar lesions sampled by serial sections at P14. (**B**) Boxplot showing statistically larger volume in P14 *Ptch⁺/⁻* lesions assigned as vascularized versus those assigned as non-vascularized, based on immunostaining serial sections of each lesion for ColIV. (**C**) Schematic illustrating the machinery required for Norrin activation of β-catenin (β-cat)/T-cell factor (TCF)-dependent transcription via the Fzd4 receptor and Lrp5 and Tspan12 co-receptors. (**D**) X-gal staining (blue) of sagittally sectioned cerebella from male *Ndp⁻/ʸ* mice carrying an *Ndp-lacZ* KO allele, at the ages indicated. Bottom row, combined X-gal staining and immunohistochemistry (IHC, brown) for phospho-histone H3 (PH3), Pax6, and myelin basic protein (MBP). (**E**) Box plot of qRT-PCR analysis of *Ndp* expression in mouse *Ptch⁺/⁻* purified GNPs and MB lysate from symptomatic animals ranging in age from 3 to 10 months. (**F**) Box plot of *NDP* expression obtained by array profiling of human cerebella and two different cohorts (Boston, left; Toronto, right) of human MB samples, categorized by molecular subgroup. (**G**) Kaplan-Meier survival curve illustrating overall survival of Shh-MB patients with high versus low *NDP* expression. EGL, external granule layer; GNP, granule neuron progenitor; PCL, Purkinje cell layer; ML, molecular layer; WM, white matter; IGL, internal granule layer; CB, cerebellum; MB, medulloblastoma. Scale bars, 100 μm.

(MBP)$^+$ white matter (*Figure 1D*) implicating oligodendrocytes as another source of Norrin. To focus our expression analysis on the *Ptch*$^{+/-}$ tumor-relevant cell type, we examined *Ndp* expression in GNPs isolated from the cerebellar surface at various stages of tumorigenesis. While *Ndp* levels in *Ptch*$^{+/-}$ GNPs from pre-lesion (P7) and early lesion (P14) stages were comparable, *Ndp* expression exhibited a downward trend in GNPs at a lesion progression stage (P30) that reached significance by the tumor stage (*Figure 1E*). In human MB, *NDP* expression is enriched specifically in the Shh subgroup compared to the other three molecularly distinct subgroups: Wnt, driven by aberrant Wnt pathway activation, and Group 3 and 4, which are less clearly defined biologically (*Taylor et al., 2012*) (*Figure 1F*). Based on a limited sample size, we observed a trend towards reduced survival in Shh-MB patients with tumors exhibiting the lowest levels of *NDP* expression compared to those with the highest (*Figure 1G*). Thus, given its known function in signalling to endothelial cells and its expression pattern in GNPs and MB, the Norrin/Fzd4 axis is well-positioned to mediate neural-endothelial cell crosstalk within the Shh-MB lesion and tumor microenvironment.

## Norrin/Fzd4 loss of function dramatically enhances *Ptch*$^{+/-}$ MB formation

To explore the relevance of the stromal compartment during Shh-MB evolution, we disrupted the Norrin/Fzd4 signalling axis at the level of the ligand or receptor in *Ptch*$^{+/-}$ mice. Given that MB incidence in *Ptch*$^{+/-}$ mice is dependent on the genetic background strain (*Mille et al., 2014*), we performed all long and short-term studies of tumorigenesis by comparing animals within the same breeding cohort. While germ-line deletion of the ligand *Ndp* (*Ndp*$^{KO}$) or conditional deletion of *Fzd4* in endothelial cells via the *Tie2Cre* driver (*Tie2Cre+;Fzd4*$^{fl/fl}$) alone did not result in tumors, both mutations dramatically accelerated MB formation and significantly reduced survival in *Ptch*$^{+/-}$ mice (*Figure 2A–C*), to an extent that has only previously been observed in this model upon DNA damage (*Pazzaglia et al., 2006a*; *2006b*; *Wetmore et al., 2001*). In both *Ndp*$^{KO}$;*Ptch*$^{+/-}$ and *Tie2-Cre+;Fzd4*$^{fl/fl}$;*Ptch*$^{+/-}$ compound mutants, MB incidence increased by approximately two-fold and mean latency was reduced more than two-fold compared to *Ptch*$^{+/-}$ littermates (*Figure 2C*). In isolated GNPs, we detected the expression of endogenous FZD4 protein (*Figure 2D*), and transcripts of *Fzd4*, *Lrp5* and *Tspan12* (*Figure 2E*); however, conditional deletion of *Fzd4* from GNPs in the *Ptch*$^{+/-}$ model had no impact on survival and tumorigenesis (*Figure 2F*). These results reveal a novel tumor inhibitory role for Norrin/Fzd4 signalling in *Ptch*$^{+/-}$ MB that, strikingly, is mediated by the endothelial cell component of the tumor stroma.

## Loss of *Ndp* alters the stromal gene expression signature in *Ptch*$^{+/-}$ MB

To gain insight into Norrin-mediated effects on *Ptch*$^{+/-}$ tumorigenesis, we examined established *Ndp*$^{KO}$;*Ptch*$^{+/-}$ and *Ptch*$^{+/-}$ MBs by histology and expression profiling. Both tumor types exhibited classic histology, and similar expression patterns for expected Shh-MB neural markers (*Atoh1*, *TUBB3* and *GFAP*) and Hh target genes (*Gli1*, *Mycn* and *Ccnd1*) (*Figure 3A*). Co-immunostaining for the ECM protein laminin and the vascular endothelial cell marker CD31 (PECAM1; platelet/endothelial cell adhesion molecule 1) revealed a stromal component in both tumors (*Figure 3B*). However, tumor heterogeneity poses challenges to histological-based classification, therefore we turned to whole genome expression profiling, where hierarchical clustering and principal component analysis revealed that *Ndp*$^{KO}$;*Ptch*$^{+/-}$ and *Ptch*$^{+/-}$ tumors have clearly separable gene expression signatures (*Figure 3C* and *Figure 3—figure supplement 1A*). Consistent with a stromal requirement for Norrin/Fzd4 signalling, tumors in *Ndp*$^{KO}$;*Ptch*$^{+/-}$ mutants were enriched for changes in stromal genes compared to their *Ptch*$^{+/-}$ counterparts, particularly ECM components (*Figure 3D,E*). Up-regulated genes in *Ndp*$^{KO}$;*Ptch*$^{+/-}$ tumors included components of endothelial cells, including *Esm1* (Endothelial cell-specific molecule 1), *Plvap* (Plasmalemmal vesicle associated protein) and *Emcn* (Endomucin) (*Figure 3—figure supplement 1C,D*), which we validated by qRT-PCR (*Figure 3F*). The up-regulation of *Plvap*, a marker of fenestrated endothelial cells, was associated with increased vascular permeability in *Ndp*$^{KO}$;*Ptch*$^{+/-}$ versus *Ptch*$^{+/-}$ tumors, demonstrated by enhanced leakage of the serum protein binding dye Evans blue (*Figure 3G*). Additional genes upregulated in *Ndp*$^{KO}$;*Ptch*$^{+/-}$ versus *Ptch*$^{+/-}$ tumors by qRT-PCR were *Pecam1*, suggesting increased vascularity in *Ndp*$^{KO}$;*Ptch*$^{+/-}$ tumors, and the angiogenic regulator *Angpt2* (Angiopoietin-2) (*Figure 3F*). These results suggest that rather than impacting Shh signalling, loss of Norrin modulates features of the tumor stroma.

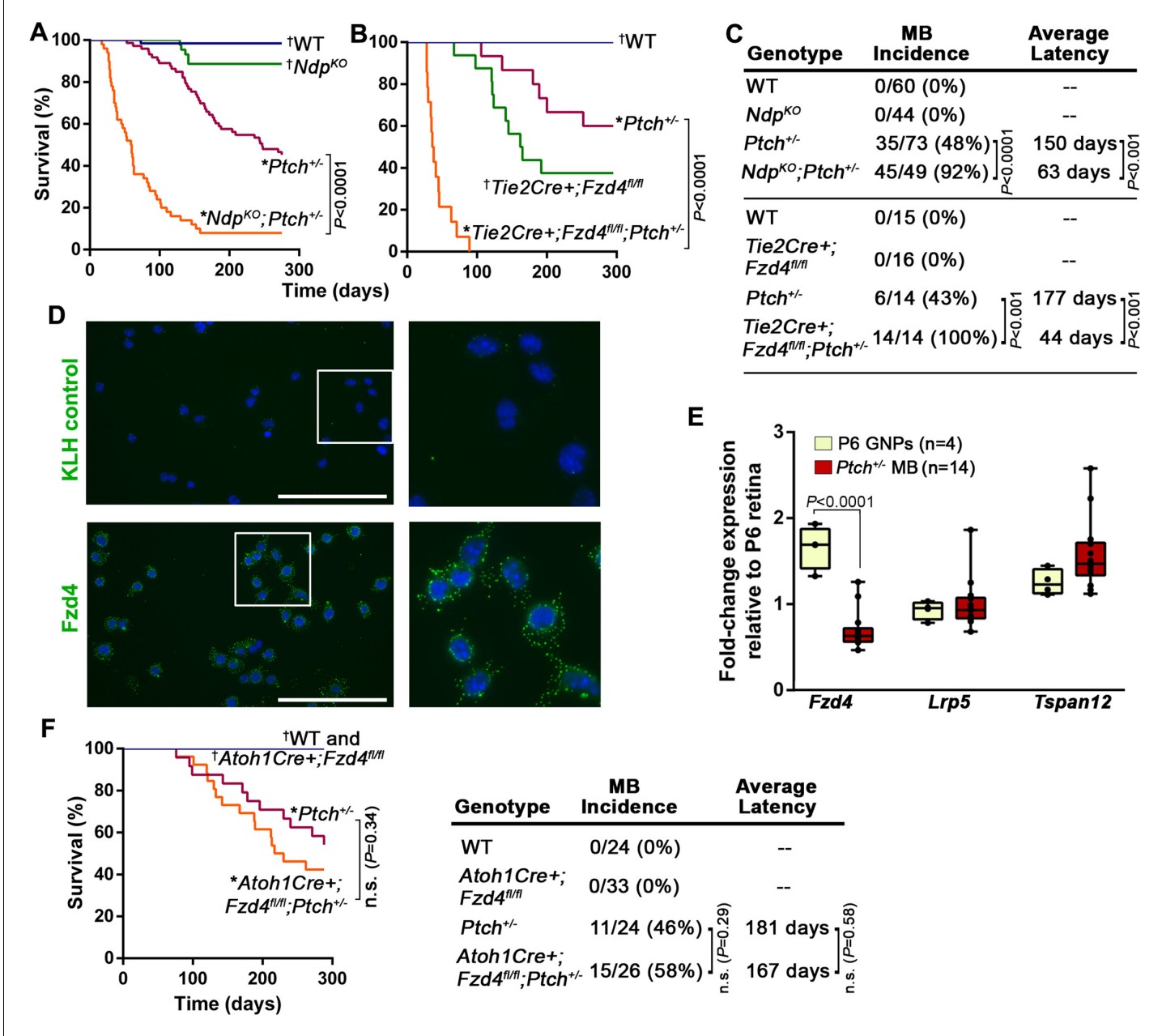

**Figure 2.** Norrin/Fzd4 signalling in endothelial cells has a potent tumor inhibitory role in *Ptch*[+/−] MB. (A) Kaplan-Meier survival curve to assess the impact of *Ndp* deletion in *Ptch*[+/−] mice. *Ndp*[KO] animals do not develop tumors but five of 44 animals were euthanized prematurely due to a skin condition. (B) Kaplan-Meier survival curve to assess the impact of endothelial cell-targeted (*Tie2Cre*-driven) *Fzd4* deletion in *Ptch*[+/−] mice. *Tie2Cre+; Fzd4*[fl/fl] animals do not develop tumors but exhibit reduced survival as reported in *Fzd4*[KO] mice, which die with esophageal-related feeding defects and progressive auditory and cerebellar degeneration (*Wang et al., 2001*). (C) Summary of sample sizes, MB incidence and average latency in all animals from Kaplan-Meier survival curves in A and B. (D) Purified P10 GNPs immunostained for anti-FZD4 and anti-keyhole limpet hemocyanin (KLH) isotype-matched control (green), counterstained with Hoescht (blue). White boxes are magnified at right. (E) Box plots of qRT-PCR analysis of Norrin receptor components in isolated mouse GNPs and *Ptch*[+/−] MB. (F) Kaplan-Meier survival curve to assess the impact of GNP-targeted (*Atoh1Cre*-driven) *Fzd4* deletion in *Ptch*[+/−] mice. (G) Summary of sample sizes, MB incidence and average latency in all animals from Kaplan-Meier survival curve in F *Died with confirmed MB; †Do not develop MB. WT, wild-type; n.s., not significant.

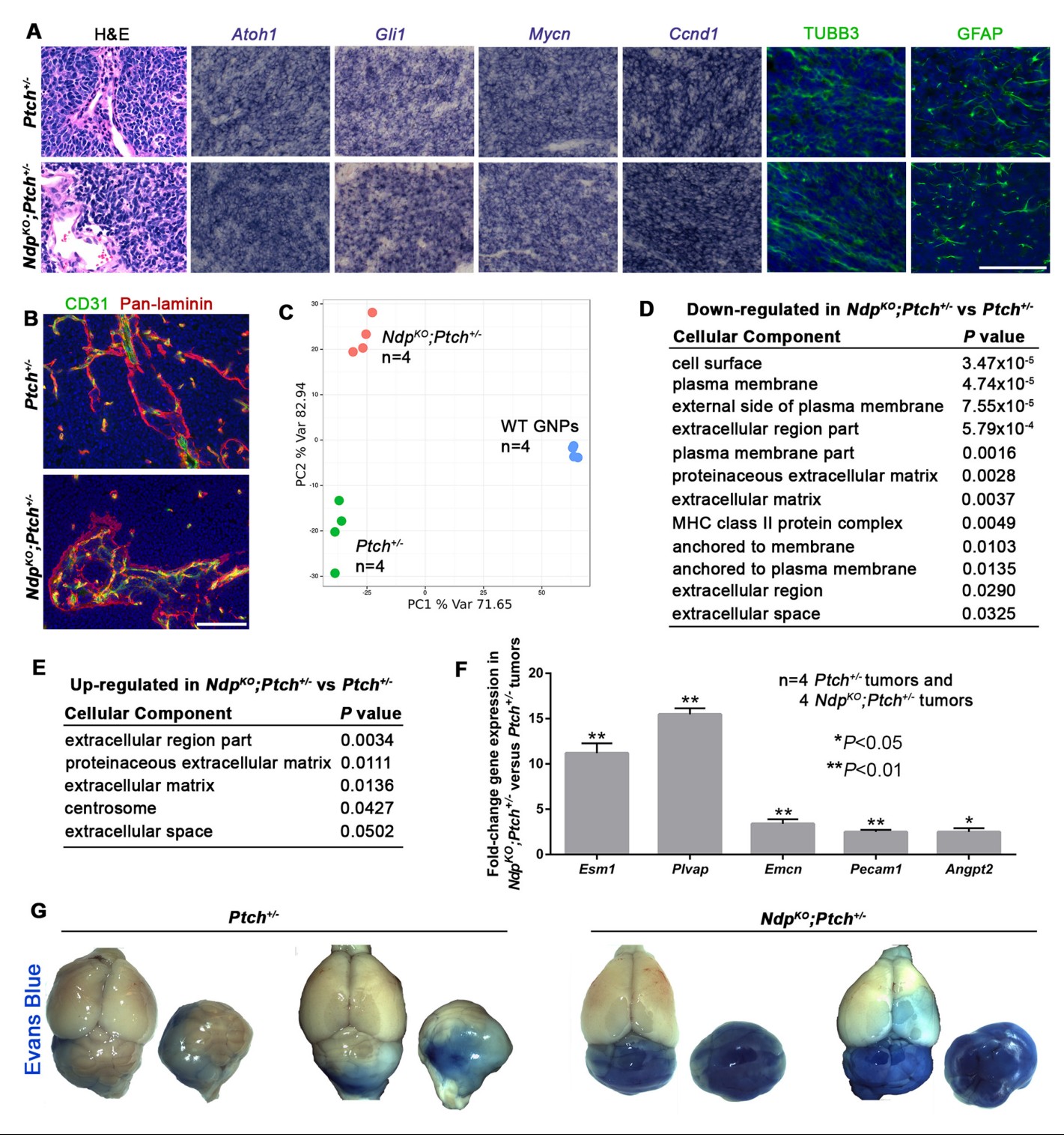

Figure 3. $Ptch^{+/-}$ and $Ndp^{KO};Ptch^{+/-}$ MBs have separable gene expression signatures enriched for stromal gene changes. (A) Sections of $Ptch^{+/-}$ and $Ndp^{KO};Ptch^{+/-}$ established MBs stained by hematoxylin and eosin (H and E), in situ hybridization for $Atoh1$, $Gli1$, $Mycn$ and $Ccnd1$ (purple), or immunostaining for class III β-tubulin (TUBB3) and glial fibrillary acidic protein (GFAP; green) counterstained with Hoescht (blue). n = 3 tumors of each genotype examined. (B) Sections of $Ptch^{+/-}$ and $Ndp^{KO};Ptch^{+/-}$ established MBs co-immunostained for CD31 (green) and pan-laminin (red), counterstained with Hoescht (blue). n = 3 tumors of each genotype examined. (C) Whole genome expression profiles of $Ptch^{+/-}$ and $Ndp^{KO};Ptch^{+/-}$ MBs and P6 WT (wild-type) GNPs were used for principal component analysis performed with the 1500 most variable probes across all samples. (D,E) Cellular component gene ontology (GO) analysis of differentially expressed genes between $Ptch^{+/-}$ and $Ndp^{KO};Ptch^{+/-}$ MBs. (F) qRT-PCR analysis of

*Figure 3 continued on next page*

*Figure 3 continued*

vascular genes upregulated in *Ndp^KO;Ptch^+/−* versus *Ptch^+/−* MBs. (G) Wholemount views of *Ptch^+/−* and *Ndp^KO;Ptch^+/−* MBs in animals injected with Evans Blue dye prior to sacrifice. Scale bars, 100 μm. See also *Figure 3—figure supplement 1*.

The following figure supplement is available for figure 3:

**Figure supplement 1.** Differential stromal gene expression in *Ptch^+/−* MB upon loss of *Ndp*.

## Stromal Norrin/Fzd4 signalling regulates lesion induction in the *Ptch^+/−* cerebellum

In the *Ptch^+/−* model, MB progression is impacted by both the number of preneoplastic lesions, and the rate at which they transform to established tumors (*Corcoran et al., 2008*; *Pazzaglia et al., 2006a*; *2006b*; *Tanori et al., 2010*). To determine the stage of tumorigenesis where stromal Norrin/Fzd4 signalling is critical, we examined lesion formation in cerebella of *Ndp^KO;Ptch^+/−* and *Tie2Cre +;Fzd4^fl/fl;Ptch^+/−* compound mutants. At P14, the earliest stage when lesions are consistently detected in *Ptch^+/−* mice, the number of cerebellar lesions in *Ndp^KO;Ptch^+/−* and *Tie2Cre+;Fzd4^fl/fl;Ptch^+/−* mutants was increased 3.9-fold and 2.4-fold, respectively, compared their *Ptch^+/−* littermates (*Figure 4A,C*). This effect on lesion number was not associated with changes to the *Ptch^+/−* cerebellum at prior ages, as we failed to detect separable GNP expression profiles or general overgrowth of the EGL in *Ndp^KO;Ptch^+/−* versus *Ptch^+/−* cerebella at P6 (*Figure 4—figure supplement 1*). Thus, we have revealed an endogenous signalling axis in the preneoplastic microenvironment that plays a protective role during the early stages of Shh-induced tumorigenesis.

## Loss of Norrin/Fzd4 signalling promotes stromal remodelling and angiogenesis in *Ptch^+/−* lesions

Together with the altered stromal expression signature in *Ndp^KO;Ptch^+/−* tumors, our finding of enhanced lesion formation suggests that loss of Norrin/Fzd4 signalling may induce stromal perturbations that impact the earliest stages of MB progression. We therefore compared stromal characteristics in lesions from *Ndp^KO;Ptch^+/−* and *Tie2Cre+;Fzd4^fl/fl;Ptch^+/−* compound mutants and their *Ptch^+/−* littermates. Consistent with our previous observations (*Figure 1A*), immunostaining for CD31 and laminin showed that the majority of *Ptch^+/−* lesions were poorly vascularized, aside from existing blood vessels associated with the meninges at the cerebellar surface (*Figure 5A*). In contrast, compound mutant lesions exhibited several hallmarks of angiogenic remodelling, including irregular deposition of ECM (*Figure 5A*), an increased frequency of mitotic endothelial cells per vessel area (*Figure 5B*), and an increase in vascular endothelial cell and laminin density (*Figure 5C*). This vascular remodelling was restricted to lesions, with only rare examples of disrupted morphology associated with the EGL in single *Tie2Cre+;Fzd4^fl/fl* cerebella (*Figure 5—figure supplement 1*). Interestingly, while the vast majority of compound mutant lesions were vascularized at P14 (78% *Ndp^KO;Ptch^+/−* and 95% *Tie2Cre+;Fzd4^fl/fl;Ptch^+/−* versus 24% *Ptch^+/−*;*Figure 5D*), compound mutant lesion volume was not increased at this early stage of tumor evolution (*Figure 4B,D*). Accordingly, we observed vascular remodeling in very small (<0.02 mm³) compound mutant lesions (*Figure 5—figure supplement 2*). Thus, vascular remodelling induced by loss of Norrin/Fzd4 is independent of the increase in lesion volume.

Consistent with the known role for Norrin/Fzd4 in normal BBB development and function [(*Wang et al., 2012*) and *Figure 6—figure supplement 1A,B*], lesions in compound mutants showed a marked loss of vascular integrity, characterized by induction of PLVAP (*Figure 6A,B*), variable loss of the tight junction protein Claudin-5 (*Figure 6A*), and leakage of Evans blue (*Figure 6C*). In contrast, single *Ptch^+/−* mice never exhibited leaky vessels (*Figure 6A–C* and *Figure 6—figure supplement 1A,B*), which is consistent with the positive regulation of BBB maintenance by active Hh signalling (*Alvarez et al., 2011*). Accordingly, the proportion of leaky lesions was significantly increased in compound mutants compared to *Ptch^+/−* (*Figure 6D*). Lesion-associated vessels in compound mutants did not show overt differences in pericyte coverage (*Figure 6—figure supplement 1D,E*), however, they did exhibit a high frequency of perivascular accumulation of CD45+ leukocytes compared with *Ptch^+/−* lesions (*Figure 6D,E*). Together, these results show that loss of Norrin/Fzd4

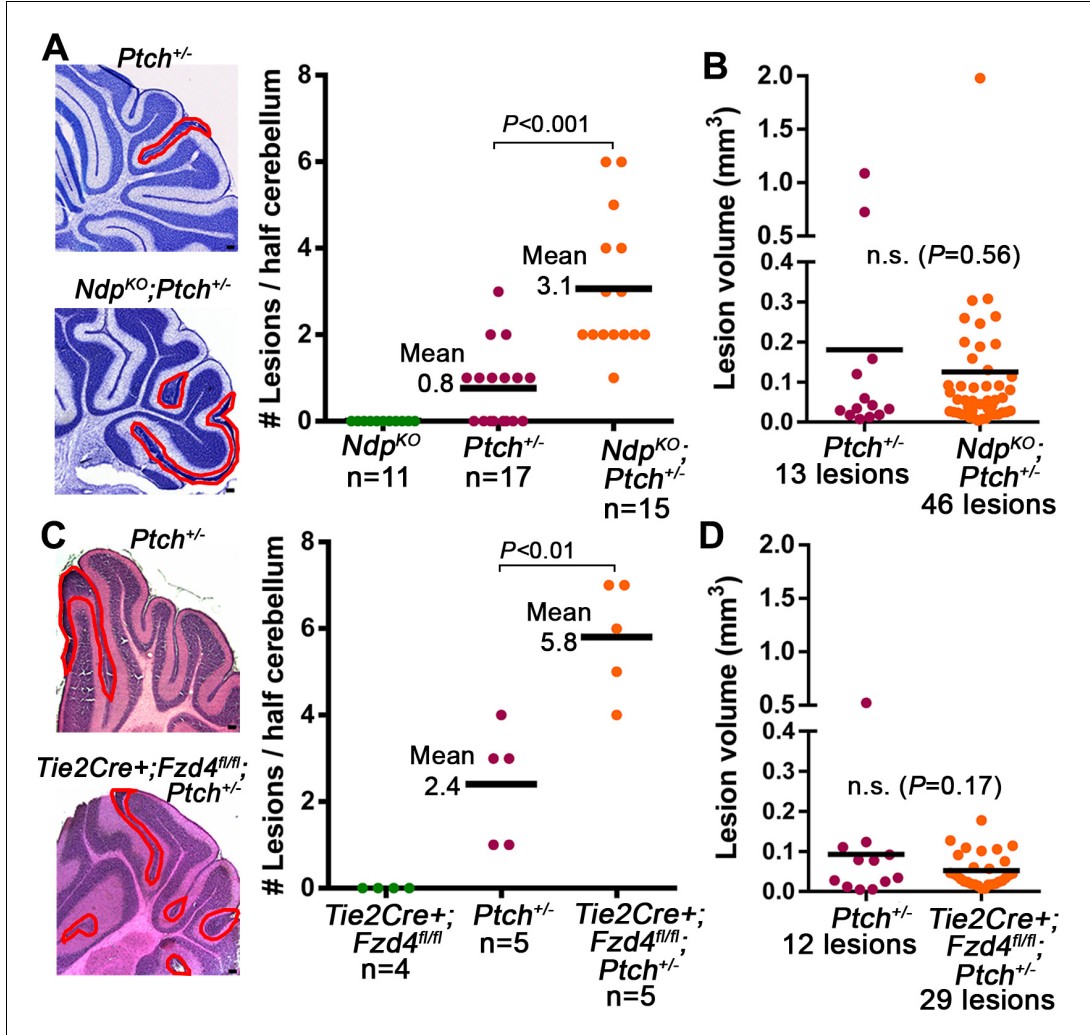

**Figure 4.** Loss of Norrin/Fzd4 signalling increases lesion formation in P14 *Ptch*$^{+/−}$ cerebella. (**A**) Quantification of lesions from serial sections of cresyl violet stained P14 cerebella from *Ndp*$^{KO}$, *Ptch*$^{+/−}$ and *Ndp*$^{KO}$;*Ptch*$^{+/−}$ mice. Example lesions are outlined in red, and n indicates the number of mice examined. (**B**) Quantification of lesion volumes from the lesions in **A**. (**C**) Quantification of lesions from serial sections of hematoxylin and eosin (H and E) stained P14 cerebella from *Tie2Cre+;Fzd4*$^{fl/fl}$, *Ptch*$^{+/−}$, and *Tie2Cre+;Fzd4*$^{fl/fl}$;*Ptch*$^{+/−}$ mice. Example lesions are outline in red, and n indicates number of mice examined. (**D**) Quantification of lesion volumes from the lesions in **C**. Means are denoted by black horizontal lines on graphs. Scale bars, 100 µm. See also *Figure 4—figure supplement 1*.

The following figure supplement is available for figure 4:

**Figure supplement 1.** Loss of Norrin signalling in *Ptch*$^{+/−}$ mice does not promote general EGL overgrowth or significant changes in GNP gene expression profile .

signalling to endothelial cells accelerates the transition to a tumor-permissive stroma characterized by vascular permeability, inflammation and angiogenic remodelling. These changes are reminiscent of the switch to angiogenesis that promotes the preneoplastic progression of non-central nervous system (CNS) tumors (*Baeriswyl and Christofori, 2009*), where the range of cellular and molecular players involved include *Angpt2* (*Mazzieri et al., 2011*; *Rigamonti et al., 2014*), consistent with our observation of increased *Angpt2* expression in *Ndp*$^{KO}$*Ptch*$^{+/−}$ tumors (*Figure 3F*).

Our findings suggest that Wnt signaling in the endothelium is normally anti-angiogenic, particularly in the context of lesion formation. Therefore, we asked whether vascularization of *Ptch*$^{+/−}$

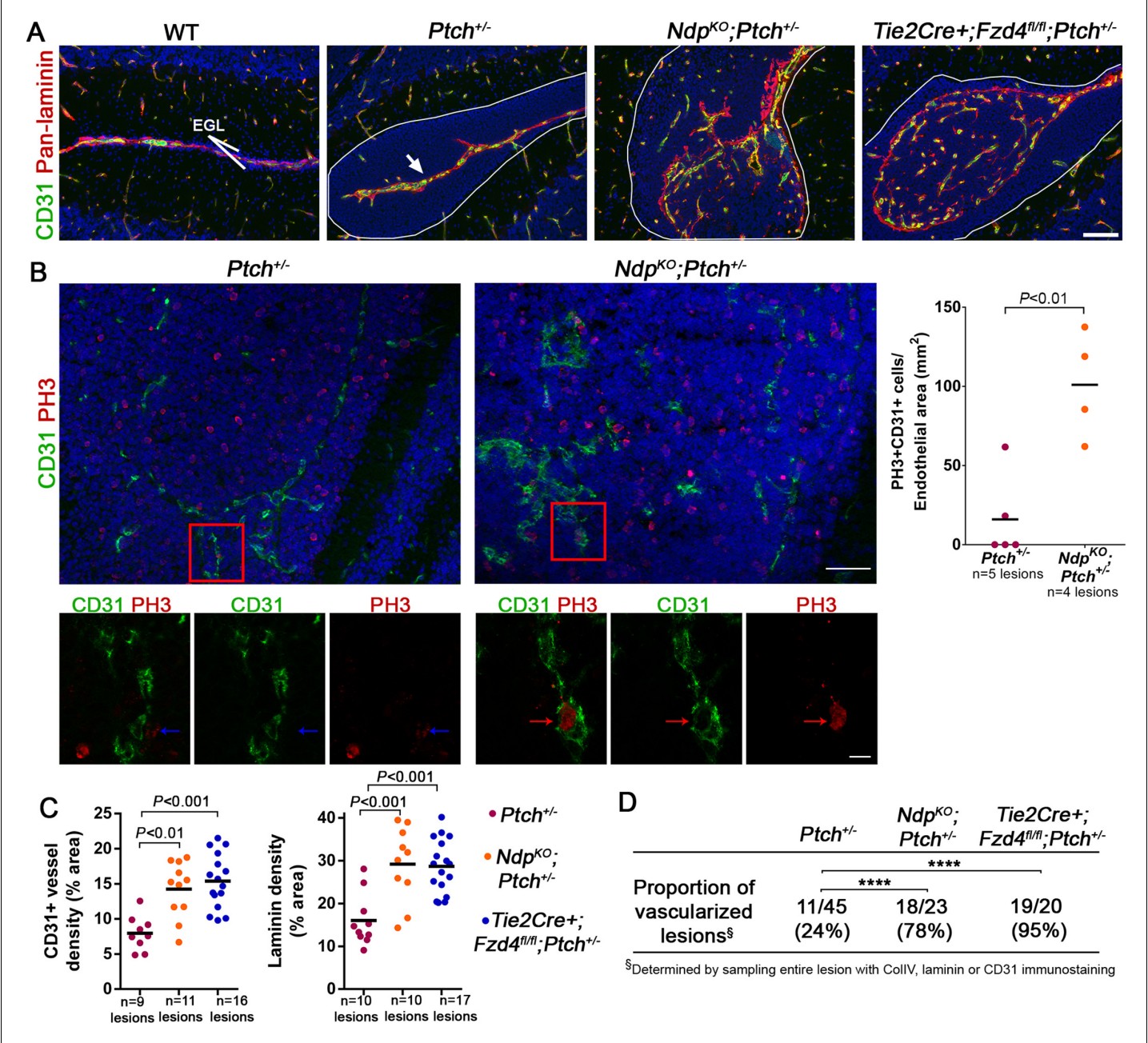

**Figure 5.** Loss of Norrin/Fzd4 signalling in endothelial cells promotes angiogenic remodeling. (**A**) Co-immunostaining for CD31 and pan-laminin, counterstained with Hoescht (blue), on sections of P14 cerebella from the genotypes indicated. Lesions are outlined in white. Arrow on *Ptch*[+/−] lesion denotes meningeal blood vessels. Scale bar, 100 μm. (**B**) Quantification of mitotic endothelial cells in *Ptch*[+/−] and *Ndp*[KO];*Ptch*[+/−] lesions. Top images show co-immunostaining for CD31 and PH3, counterstained with Hoescht (blue), on vascularized P14 lesion sections. Scale bar, 50 μm. Red squares denote areas shown by confocal scans below, where left image depicts the composite maximum intensity projection and the center and right images show individual z-stack slices. Scale bar, 10 μm. Blue arrows denote a PH3+ cell scored as negative for co-localization, whereas red arrows denote a positive co-localization. Graph on right summarizes quantification of double labelled PH3+CD31+ cells per endothelial area. (**C**) Quantification of CD31 + vessel density and laminin density in lesions of *Ptch*[+/−], *Ndp*[KO];*Ptch*[+/−] and *Tie2Cre+;Fzd4*[fl/fl];*Ptch*[+/−]. Number of lesions (n) examined is indicated on each graph in **B** and **C**, and means are denoted by black horizontal lines. (**D**) Summary of the proportion of vascularized lesions from each genotype. ****p<0.0001. See also *Figure 5—figure supplement 1* and *2*.

The following figure supplements are available for figure 5:

**Figure supplement 1.** EGL-associated morphology in Norrin/Fzd4-deficient cerebella .

*Figure 5 continued on next page*

*Figure 5 continued*

**Figure supplement 2.** Lesion size is not correlated with vascular remodeling in $Ndp^{KO};Ptch^{+/-}$ and $Tie2Cre+;Fzd4^{fl/fl};Ptch^{+/-}$ compound mutants.

lesions was associated with altered Wnt signalling by examining expression of Lef1, a canonical Wnt target gene (*Logan and Nusse, 2004*). Endothelial Lef1 expression in P14 $Ndp^{KO};Ptch^{+/-}$ lesions was significantly reduced compared to P14 $Ptch^{+/-}$ lesions and EGL (*Figure 6—figure supplement 2A,B*), confirming the *Ndp* dependence of this marker in the endothelium. While Lef1 expression in the endothelium of non-vascularized $Ptch^{+/-}$ lesions was similar to that in adjacent EGL, we observed a downregulation of endothelial Lef1 expression in vascularized $Ptch^{+/-}$ lesions compared to EGL (*Figure 6—figure supplement 2B*). These data suggest that lesion progression in $Ptch^{+/-}$ mice may be associated with a reduction in endogenous canonical Wnt activity in the endothelium, which is also consistent with the downward trend in *Ndp* expression during $Ptch^{+/-}$ lesion progression.

## The tumor-protective microenvironment has an acute requirement for Fzd4 activity

Genetic deletion of *Ndp* or *Fzd4* impacts cerebellar vasculature throughout development and adulthood (*Wang et al., 2012*; *Zhou et al., 2014*). Thus, we determined whether we could recapitulate the effects of genetic inactivation on the development of a tumor-permissive stroma by short-term treatment of $Ptch^{+/-}$ mice with a function blocking Fzd4 antibody (αFzd4), which has been previously shown to effectively cross the BBB and impact retinal vasculature (*Paes et al., 2011*) (*Figure 7—figure supplement 1*). αFzd4 treatment of $Ptch^{+/-}$ mice at P7 and P16 was sufficient to phenocopy the enhanced tumor initiation phenotype of our compound mutant models, increasing MB incidence from 69% in animals injected with anti-keyhole limpet hemocyanin (αKLH) control antibody to 93% in those injected with αFzd4, and significantly reducing mean tumor latency from 167 to 104 days (*Figure 7A*). Furthermore, a single dose of αFzd4 administered as late as P7 was sufficient to increase number (but not volume) of lesions at P14 (*Figure 7B*), convert lesions to a leaky vasculature phenotype (*Figure 7C,I*), and significantly increase the proportion of vascularized lesions compared to αKLH controls (*Figure 7D,I*). Thus, acute disruption of Fzd4 promotes the conversion to an angiogenic, pro-tumor stroma within seven days, suggesting that continued maintenance of Norrin/Fzd4 signalling creates a tumor-protective microenvironment.

## Ptx-Induced BBB disruption does not affect $Ptch^{+/-}$ tumorigenesis

To further examine what feature of the Norrin/Fzd4-deficient vasculature is pro-tumorigenic, we teased apart the impact of vessel permeability alone using pertussis toxin (Ptx), a stimulator of mouse BBB disruption (*Clifford et al., 2007*). Ptx treatment compromised paracellular permeability in the P14 cerebellum by disrupting endothelial cell tight junctions via reduced ZO-1 and Claudin-5 expression, accompanied by Evans Blue leakage (*Figure 7—figure supplement 2A,B* and data not shown). This Ptx-mediated effect on BBB integrity was independent of PLVAP induction (*Figure 7—figure supplement 2C*), an indicator of transcellular openings (*Stan et al., 2004*). Ptx treatment of $Ptch^{+/-}$ mice did not affect MB incidence or latency compared to $Ptch^{+/-}$ PBS-injected controls (*Figure 7E*). Furthermore, despite vascular permeability as seen by Evans Blue leakage in lesions of Ptx-treated $Ptch^{+/-}$ mice, the number, volume and proportion of vascularized lesions were not changed (*Figure 7F–I*). Although Ptx has been shown in certain contexts to inhibit Hh signalling by disrupting Gαi protein coupling to Smoothened (*Riobo et al., 2006*), our failure to observe changes in lesion number or MB latency suggests that GNP proliferation was not suppressed in this system. These data show that Ptx-induced BBB disruption alone does not drive lesion formation, lesion angiogenesis, or tumor initiation.

## Loss of Norrin signalling accelerates *Ptch* LOH in $Ptch^{+/-}$ MB

The striking acceleration of $Ptch^{+/-}$ MB initiation observed upon loss of Norrin/Fzd4 activity (*Figure 2A–C*) prompted us to assess indicators of GNP growth and malignant progression. We first observed a modest, but significant, increase in the mitotic index of P14 lesions of $Ndp^{KO};Ptch^{+/-}$ mutants compared to their $Ptch^{+/-}$ counterparts (*Figure 8A*), which was not counterbalanced by

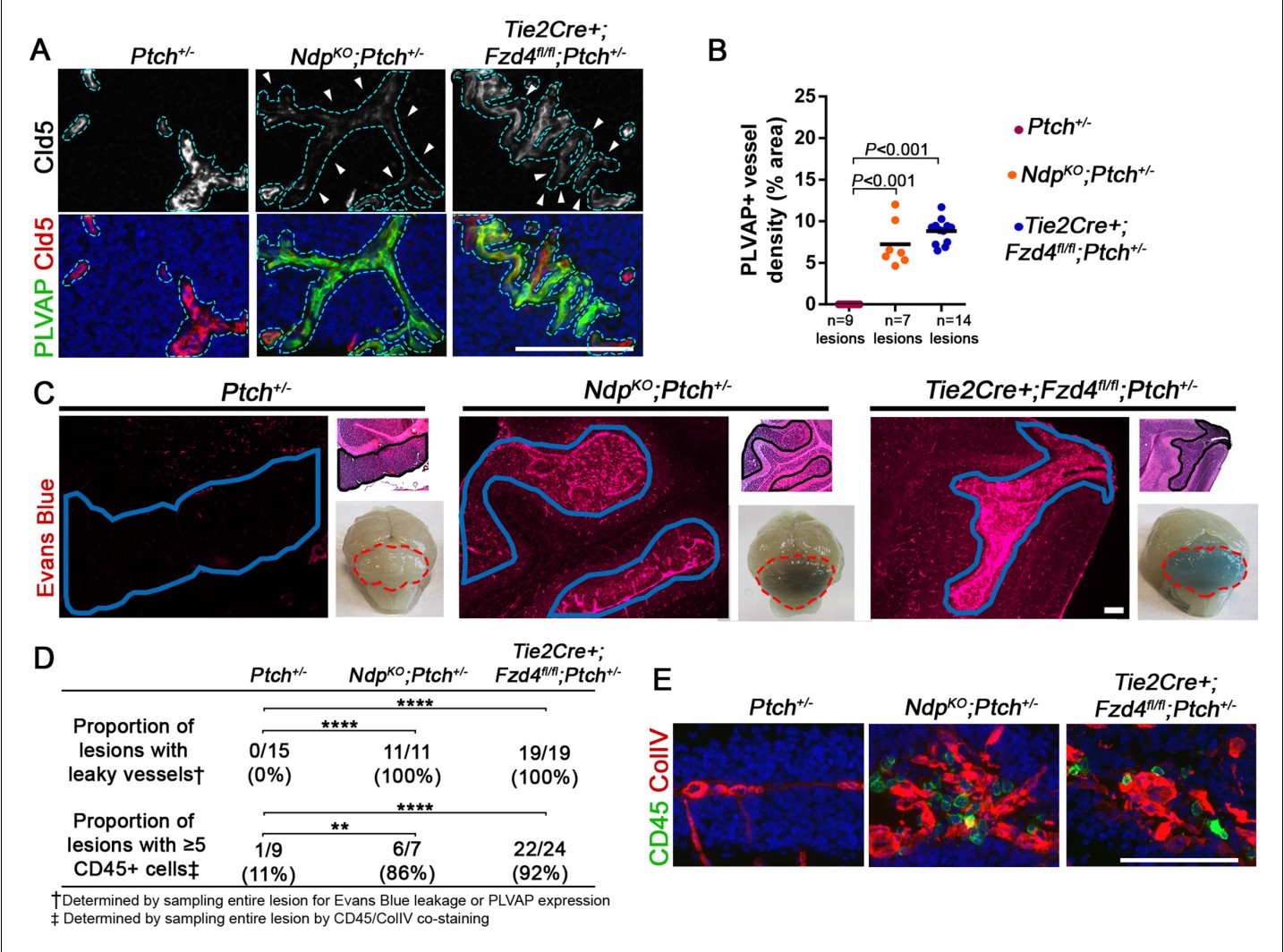

**Figure 6.** Loss of Norrin/Fzd4 signalling in endothelial cells promotes vessel leakiness in *Ptch*$^{+/-}$ lesions. (**A**) Co-immunostaining for PLVAP and Claudin-5 (Cld5) counterstained with Hoescht (blue), on lesion sections of P14 cerebella from the genotypes indicated. Vessels outlined in turquoise. Top images show Cld5 channel only, to illustrate variable reduction in Cld5 expression (arrowheads). (**B**) Quantification of PLVAP+ vessel density in lesions of *Ptch*$^{+/-}$, *Ndp*$^{KO}$;*Ptch*$^{+/-}$ and *Tie2Cre+;Fzd4*$^{fl/fl}$;*Ptch*$^{+/-}$. Number of lesions (n) examined is indicated on each graph, and means are denoted by black horizontal lines. (**C**) P14 *Ptch*$^{+/-}$, *Ndp*$^{KO}$;*Ptch*$^{+/-}$ and *Tie2Cre+;Fzd4*$^{fl/fl}$;*Ptch*$^{+/-}$ mice injected with Evans Blue dye prior to sacrifice. Sections containing lesions (outlined in blue) show Evans Blue as red fluorescence, followed by adjacent H and E-stained sections and whole brain images (cerebella outlined in red). (**D**) Summary of the proportion of lesions from each genotype containing leaky vessels or ≥5 CD45+ cells. ****p<0.0001; **p<0.01. (**E**) Co-immunostaining for CD45 and Collagen IV (ColIV) on lesion sections of P14 cerebella. Scale bars, 100 µm. See also *Figure 6—figure supplement 1* and *2*.

The following figure supplements are available for figure 6:

**Figure supplement 1.** Norrin/Fzd4-mediated cerebellar vascular defects .

**Figure supplement 2.** Vascularization of *Ptch*$^{+/-}$and *Ndp*$^{KO}$;*Ptch*$^{+/-}$ lesions is associated with reduced expression of the Wnt target Lef1 in endothelial cells .

enhanced cell death or differentiation, as *Ndp*$^{KO}$;*Ptch*$^{+/-}$ lesions exhibited less Cleaved Caspase 3 immunostaining and comparable expression levels of the differentiation marker NeuN compared to their *Ptch*$^{+/-}$ littermates (*Figure 8A*). Interestingly, this increased mitotic index did not translate into increased lesion volume at this stage (*Figure 4B,D*). We examined the relationship between

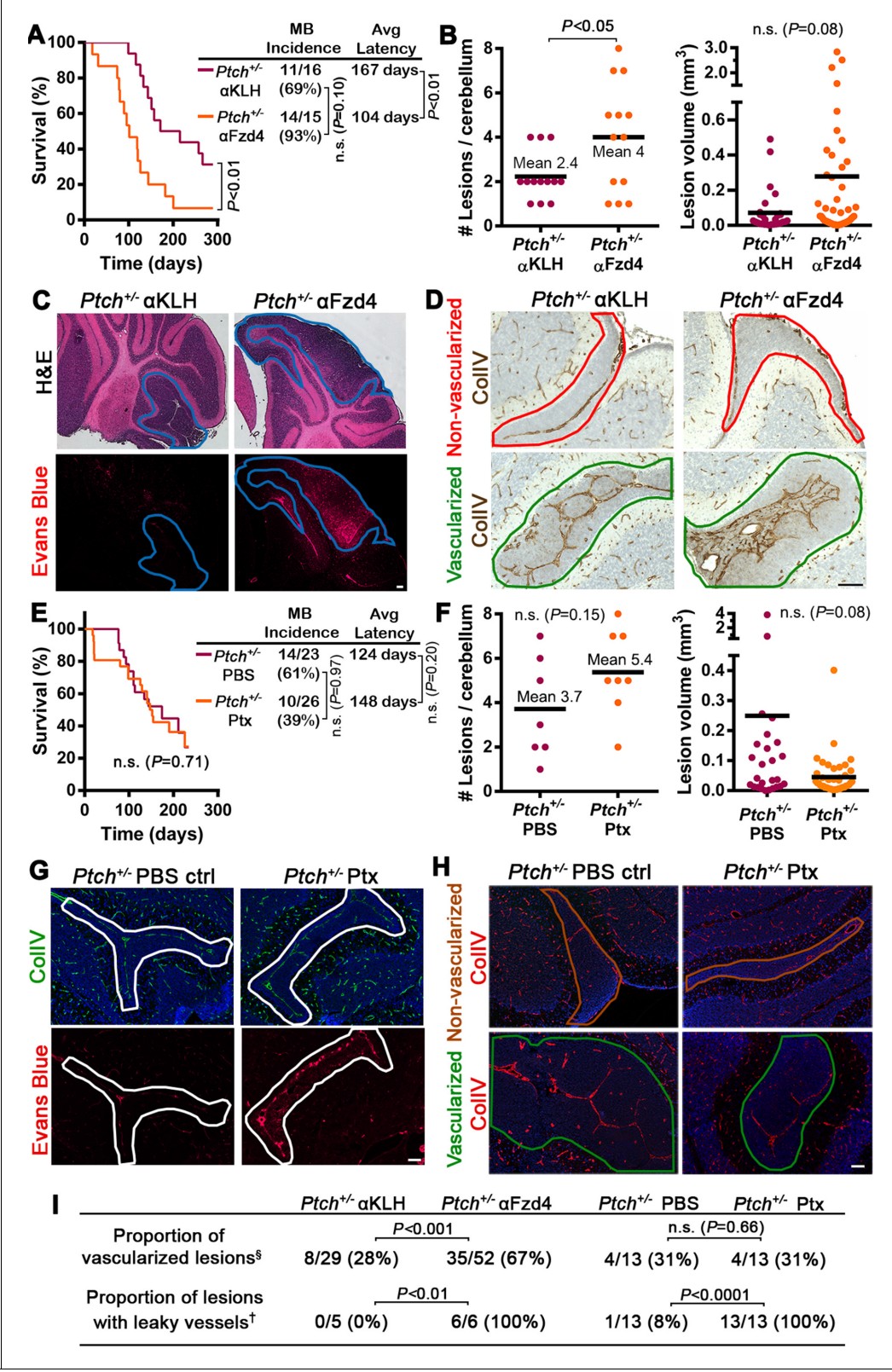

**Figure 7.** Acute disruption of Fzd4 promotes the conversion to a pro-tumor stroma . (**A**) Kaplan-Meier survival curve comparing *Ptch*[+/−] mice treated with αFzd4 or αKLH isotype matched control antibodies at P7 and P16. All mice died with confirmed MB. Sample sizes, MB incidence and average latency are indicated at right. (**B**) Quantification of lesion number and volume from P14 cerebella of *Ptch*[+/−] mice treated at P7 with αFzd4 (*n* = 13 mice, 52 lesions total) or αKLH (*n* = 13 mice, 29 lesions total). Means are denoted by black horizontal lines on graphs. (**C**) Lesion images from *Ptch*[+/−] mice

*Figure 7 continued on next page*

*Figure 7 continued*

treated at P7 with αKLH (*n* = 5 lesions) or αFzd4 (*n* = 6 lesions). Mice were injected with Evans Blue dye prior to sacrifice and sampled by H and E staining followed by Evans Blue visualization as red fluorescence on adjacent sections. Lesions outlined in blue. (**D**) Lesion images from *Ptch*[+/−] mice treated at P7 with αKLH (*n* = 29 lesions) or αFzd4 (*n* = 52 lesions), immunostained for ColIV and counterstained with hematoxylin, to quantify the proportions of non-vascularized (outlined in red) and vascularized (outlined in green) lesions in each group. (**E**) Kaplan-Meier survival curve comparing *Ptch*[+/−] mice treated with Ptx or PBS vehicle control at P7, P9, P11 and P13. The sudden drop in Ptx-treated survival is a result of four animals dying from brain hemorrhages or seizures. Three other Ptx-treated animals were euthanized due to malocclusion or unknown causes, while all other mice in both groups died with confirmed MB. Sample sizes, MB incidence and average latency are indicated at the right. (**F**) Quantification of lesion number and volume from P14 cerebella of *Ptch*[+/−] mice treated as above with Ptx (*n* = 8 mice, 43 lesions total) or PBS (*n* = 7 mice, 26 lesions total). (**G**) Lesion images from *Ptch*[+/−] mice treated with Ptx (*n* = 13 lesions) or PBS (*n* = 13 lesions) as above. Mice were injected with Evans Blue dye prior to sacrifice and sampled by ColIV immunostaining followed by Evans Blue visualization as red fluorescence on adjacent sections. Lesions outlined in white. (**H**) Lesion images from *Ptch*[+/−] mice treated with Ptx (*n* = 13 lesions) or PBS (*n* = 13 lesions) as above, immunostained for ColIV to quantify the proportions of non-vascularized (outlined in brown) and vascularized (outlined in green) lesions in each group. (**I**) Summary of *Ptch*[+/−] lesion number and vessel parameters upon treatment with αFzd4 or Ptx. Scale bars, 100 μm. See also *Figure 7—figure supplement 1* and *2*.

The following figure supplements are available for figure 7:

**Figure supplement 1.** Functional in vivo validation of anti-Fzd4 blocking antibody .

**Figure supplement 2.** Disruption of cerebellar endothelial cell tight junctions upon treatment with pertussis toxin (Ptx) .

proliferating GNPs and lesion vasculature by Pax6/CD31/EdU triple staining, and found that Pax6 +EdU+ S-phase GNPs were present in the vicinity of blood vessels in *Ptch*[+/−] or *Ndp*[KO];*Ptch*[+/−] lesions (*Figure 8—figure supplement 1*). In the *Ptch*[+/−] MB context, enhanced proliferation and DNA damage in pre-malignant GNPs leads to increased incidence and decreased latency of MB (*Ayrault et al., 2009*; *Leonard et al., 2008*; *Mille et al., 2014*; *Uziel et al., 2005*). Here, *Ptch* loss of heterozygosity (LOH; where the remaining wild-type *Ptch* allele is inactivated) is a well-established indicator of malignant progression in this model (*Mille et al., 2014*; *Pazzaglia et al., 2006a*; *2006b*). To determine if Norrin/Fzd4 signalling affects the rate of *Ptch* LOH, we microdissected GNPs from lesions in *Ptch*[+/−] and *Ndp*[KO];*Ptch*[+/−] cerebella at P14, and determined *Ptch* LOH status by detection of wild-type *Ptch* transcript (*Figure 8B*). At this stage, the frequency of *Ptch* LOH was increased in *Ndp*[KO];*Ptch*[+/−] lesions (9/17) compared to *Ptch*[+/−] lesions (3/14). Thus, the shorter MB latency observed upon *Ndp* loss-of-function is characterized by an accelerated rate of *Ptch* LOH.

To address whether the tumor inhibitory effect of Norrin/Fzd4 signaling extends to models of Shh-MB that do not require *Ptch* LOH for progression, we used the *Neurod2-Smo*[A1] transgenic mouse, a Shh-MB model where an oncogenic form of Smoothened with an activating point mutation SmoA1 is expressed in GNPs (*Hallahan et al., 2004*). Deletion of *Ndp* dramatically accelerated cerebellar tumorigenesis in *Neurod2-Smo*[A1+/−] mice, reducing mean latency almost 5-fold, from 167 to 34 days (*Figure 8C*). Thus, the tumor inhibitory effects of Norrin signaling extend to oncogene-driven Shh MB.

## Discussion

### A novel link between stromal signalling and MB initiation

Stromal elements are increasingly recognized as critical to the initiation and progression of many tumor types, however little is known about the contribution of the microenvironment to MB evolution. By exploiting the multi-step nature of tumorigenesis in the *Ptch*[+/−] mouse model, we have described a novel stromal interaction regulating *Ptch*[+/−] MB initiation and identified a previously unrecognized angiogenic component to the earliest tumorigenic events in the MB microenvironment (*Figure 8D*). We uncovered a cellular mechanism behind this effect, identifying endothelial cells as the mediators of a potent tumor inhibitory signal that acts during preneoplasia and oncogenic transformation. Furthermore, we identified the Norrin/Fzd4 pathway as an endogenous signalling axis that can be manipulated to target this pre-tumor/stromal relationship. The importance of Wnt signalling in brain tumor vasculature has been demonstrated in established human glioma, where tumor cell-derived Wnt ligand normalizes vasculature and inhibits angiogenesis (*Reis et al., 2012*), and in

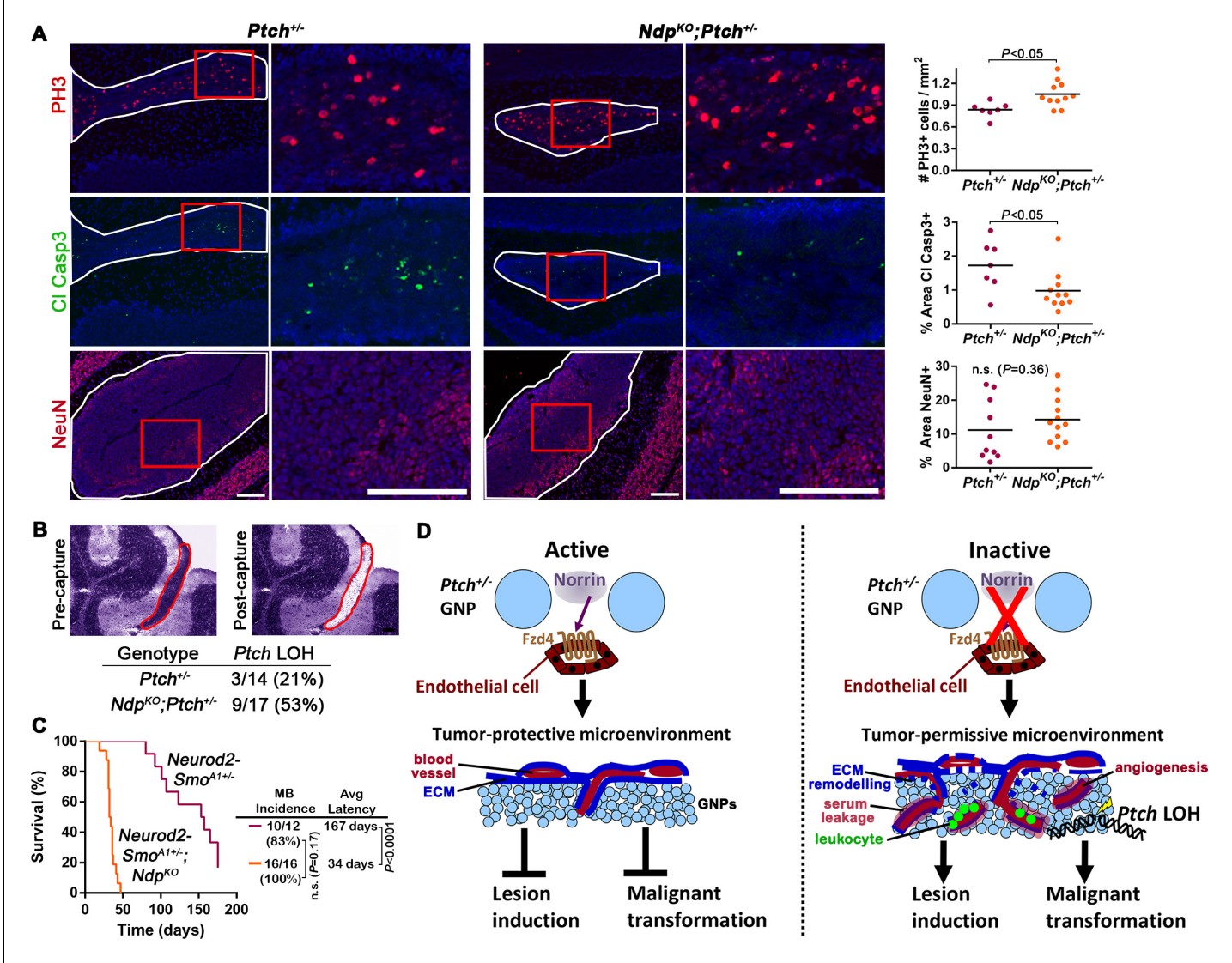

**Figure 8.** Loss of Norrin signalling accelerates the transition to malignancy in $Ptch^{+/-}$ lesions. (**A**) Immunostaining and quantification of proliferation (PH3; $n=7$ $Ptch^{+/-}$ lesions and $n=11$ $Ndp^{KO};Ptch^{+/-}$ lesions), apoptosis (cleaved caspase 3; $n=7$ $Ptch^{+/-}$ lesions and $n=11$ $Ndp^{KO};Ptch^{+/-}$ lesions) and differentiation (neuronal nuclear protein NeuN; $n=10$ $Ptch^{+/-}$ lesions and $n=12$ $Ndp^{KO};Ptch^{+/-}$ lesions) on P14 cerebellar lesions (outlined in white) counterstained with Hoescht (blue). Areas in red boxes are magnified at right. Means are denoted by black horizontal lines on graphs. (**B**) Frequency of $Ptch$ loss of heterozygosity (LOH) in lesions of P14 $Ptch^{+/-}$ and $Ndp^{KO};Ptch^{+/-}$ mice, determined by wild-type allele-specific detection of $Ptch$ transcripts from microdissected lesions. Example images depict a toluidine blue-stained cerebellar section pre- and post-laser capture (lesion outlined in red). (**C**) Kaplan-Meier survival curve to assess the impact of $Ndp$ deletion in the $Neurod2$-$Smo^{A1+/-}$ model of MB (**D**) Schematic to illustrate the effects of Norrin/Fzd4 signalling on $Ptch^{+/-}$ MB progression. n.s., not significant. ECM, extracellular matrix. Scale bars, 100 μm. See also **Figure 8—figure supplement 1**.

The following figure supplement is available for figure 8:

**Figure supplement 1.** Proliferating GNPs are present in the vicinity of blood vessels in $Ptch^{+/-}$ and $Ndp^{KO};Ptch^{+/-}$ lesions .

Wnt-MB, where tumor cells secrete Wnt inhibitors to disrupt BBB stability (**Phoenix et al., 2016**). Here we have extended the concept of tumor-stroma crosstalk via Wnt signalling to the earliest stages of tumorigenesis in the brain.

## Norrin/Fzd4 signalling and human tumor progression

Mutations in Norrin/Fzd4 pathway components cause a spectrum of human pathologies, including vascular disorders with retinal and cerebellar phenotypes (*Gilmour, 2015*; *Liu et al., 2010*; *Romaniello et al., 2013*). However, the role of this signalling axis in the vasculature has not been explored in human tumorigenesis. Based on our data, one might predict an inverse relationship between *NDP* levels and outcome in human Shh-MB (such as earlier age of onset or reduced survival). While we did observe a trend towards reduced survival in Shh-MB patients with the lowest levels of *NDP* transcript, *NDP* mRNA levels are generally high in human Shh-MB, suggesting a potential protective role for this pathway. However, transcript abundance may not be a universal indicator of Norrin pathway activation. Notably, the activities of several angiogenic factors, including VEGF and FGF, are regulated at the level of protein bioavailability (*Hynes, 2009*), requiring release from the ECM to mediate angiogenesis in pre-tumor lesions (*Bergers et al., 2000*; *Nozawa et al., 2006*). Interestingly, Norrin also associates with the ECM (*Perez-Vilar and Hill, 1997*). We also note that, given the clonal evolution patterns resulting in tumor heterogeneity within established mouse and human Shh-MB (*Wu et al., 2012*), early initiation events will not necessarily be reflected in expression profiles of the evolved tumors in patient tumor banks. Nonetheless, given the known role of Wnt-mediated neural/endothelial crosstalk in CNS vascular development and pathology (*Engelhardt and Liebner, 2014*; *Phoenix et al., 2016*; *Ye et al., 2010b*), our work highlights the importance of further examining the Norrin/Fzd4 pathway in the human MB progression, including a potential prophylactic role in susceptible individuals.

## The preneoplastic niche and *Ptch*$^{+/-}$ MB initiation

*Ptch*$^{+/-}$ MB incidence and latency can be altered by modulating several pathways involved in proliferation and DNA damage (*Ayrault et al., 2009*; *Briggs et al., 2008*; *Uziel et al., 2005*; *Wetmore et al., 2001*); however there are few factors known to conclusively impact the number of preneoplastic lesions or their transition to malignancy. *Ptch*$^{+/-}$ lesion induction is known to be mediated by growth or differentiation regulators [*Igf1*, *Ccnd1*, *Tis21* (*Farioli-Vecchioli et al., 2012*; *Pogoriler et al., 2006*; *Tanori et al., 2010*)] and by radiation or interference with DNA repair (*Malek et al., 2011*; *Pazzaglia et al., 2006a*; *2006b*), whereas progression of lesions to malignancy is enhanced by the growth promoting factors *Igf2*, *Mycn*, Shh co-receptor *Boc* and by the evasion of p53-induced senescence (*Corcoran et al., 2008*; *Kessler et al., 2009*; *Mille et al., 2014*; *Tamayo-Orrego et al., 2016*). Here, we introduce the concept that a permissive preneoplastic microenvironment will significantly enhance *Ptch*$^{+/-}$ lesion induction and tumor initiation. We have linked the pro-tumor stroma induced by loss of Norrin/Fzd4 signalling to increased GNP proliferation and accelerated *Ptch* LOH, both events that are known to increase the incidence and decrease latency of *Ptch*$^{+/-}$ MB (*Ayrault et al., 2009*; *Mille et al., 2014*; *Pazzaglia et al., 2006a*; *2006b*; *Uziel et al., 2005*). These data add to accumulating evidence that development of an oncogenic niche supports the growth and transformation of preneoplastic cells (*Barcellos-Hoff et al., 2013*). Although beyond the scope of our study to investigate in detail, the persistence of stromal changes in *Ndp*$^{KO}$;*Ptch*$^{+/-}$ established tumors highlights the relevance of examining the ongoing role of tumor-stroma crosstalk in the growth of MB.

## Blood vessels as mediators of tumorigenesis in the *Ptch*$^{+/-}$ cerebellum

We have identified several Norrin/Fzd4-dependent stromal abnormalities associated with MB initiation, and our data suggest that disrupting the paracellular barrier property of the endothelium alone is not causal. Similarly, a recent study showed that modulation of Wnt-mediated BBB characteristics in orthotopic transplants of established mouse MB did not impact the tumor incidence or survival (*Phoenix et al., 2016*). We recognize that *Ptch*$^{+/-}$ vascular endothelial cells in the germ-line *Ptch*$^{+/-}$ model may exhibit elevated Hh pathway activity; however this is a positive regulator of BBB formation (*Alvarez et al., 2011*). Furthermore, our data showing dramatic acceleration of tumor formation upon deletion of *Ndp* from the *Neurod2-Smo*$^{A1}$ MB model demonstrate that Norrin-dependent effects on tumorigenesis are independent of *Ptch* status in the endothelium. We also note that while Ptx treatment disrupted junctional protein expression in the cerebellar endothelium, this was not accompanied by induction of PLVAP, therefore we cannot rule out transcellular permeability as a

possible feature of the pro-tumor stroma, or the possibility that the severity of BBB disruption is important.

Our work suggests that attenuation of Wnt signaling in endothelial cells and vascular invasion are endogenous factors regulating $Ptch^{+/-}$ lesion progression. These data are consistent with the finding that, although Norrin activity is not fully absent in $Ptch^{+/-}$ lesions (indicated by a largely intact BBB) $Ndp$ expression is downregulated as a function of tumorigenesis. The exact mechanism by which endothelial cells deficient in Norrin/Fzd4 signalling could alter the $Ptch^{+/-}$ preneoplastic niche and affect GNPs remains to be determined. In $Tie2Cre+;Fzd4^{fl/fl}$ mutants, we detected rare foci of disrupted morphology involving EGL-associated vessels, suggesting that surface vasculature adjacent to the EGL is particularly susceptible to Norrin-mediated instability. Thus, the interaction between $Ptch^{+/-}$ GNPs and endothelial cells lacking Norrin/Fzd4 pathway activation may provide a unique setting for lesion growth and vascular invasion. Interestingly, previous work analyzing the transcriptomes of $Fzd4^{KO}$ versus $Fzd4$ WT endothelial cells isolated from the adult mouse retina and P16 cerebellum (*Ye et al., 2009*) showed that $Fzd4$-deficient endothelial cells exhibit transcript enrichment for genes known to promote both GNP growth ($Igf1$) and angiogenesis ($Angpt2$ and $Esm1$, both of which were upregulated in $Ndp^{KO};Ptch^{+/-}$ versus $Ptch^{+/-}$ MB). However, we have not ruled out the possibility that inflammation is involved. The $Tie2Cre$ driver used to delete $Fzd4$ is also expressed in hematopoietic lineages (*Tang et al., 2010*), which, combined with our observation of leukocyte accumulation in $Ndp/Fzd4$-deficient lesions, leaves open the prospect of an immune cell-mediated component within the pro-tumor stroma.

## The permissive preneoplastic niche: A recurring feature of multistage tumor models

In tumors with defined stages of progression, such as pancreatic and breast cancer, the interaction between preneoplastic cells and their surrounding stroma is recognized as a critical modulator of tumor evolution (*Coussens et al., 1996*; *Gullino, 1978*; *Hanahan and Folkman, 1996*; *Lin et al., 2006*). Mouse models of these tumors have shown that the angiogenic switch, in particular, occurs prior to malignancy and may be a rate limiting step for tumor initiation (*Giraudo et al., 2004*; *Hanahan et al., 1996*; *Lin et al., 2006*; *Smith-McCune et al., 1997*). Here, we examined cerebellar lesions at the earliest stage that they are consistently detected in the $Ptch^{+/-}$ cerebellum, P14. At this stage, although we detected a modest increase in proliferation and accelerated $Ptch$ LOH in $Ndp^{KO};Ptch^{+/-}$ mutants, we did not find that disruption of Norrin/Fzd4 activity was associated with an increase in lesion volume. These data reflect previous studies in pancreatic and breast tumor models, where the angiogenic switch was not correlated with size in early-stage lesions, but was instead associated with, and required for, the transition from preneoplasia to malignancy (*Folkman et al., 1989*; *Lin et al., 2006*). Following this transition, transformed lesions will clearly have a growth advantage. The mechanism by which blood vessels in the pre-tumor microenvironment can promote transformation is not defined. Undoubtedly, further delineating the molecular events mediating crosstalk between preneoplastic cells and stromal elements will be important for understanding tumorigenesis in multiple cancers.

## Materials and methods

### Mice

All experiments were approved by the University of Ottawa Animal Care Ethics Committee and adhered to the guidelines of the Canadian Council on Animal Care (CCAC). $Ndp^{KO}$ mice (RRID:MGI: 4414648), generated by disruption of the $Ndp$ locus by a $lacZ$-containing cassette were obtained from Lexicon Pharmaceuticals (*Junge et al., 2009*) and maintained by interbreeding on a mixed background. $Ndp$ is an X-linked gene, therefore $Ndp^{KO}$ males and females have $Ndp^{-/y}$ and $Ndp^{-/-}$ genotypes, respectively. $Ndp^{-/y}$ males were used for $lacZ$-based reporter expression analysis in *Figure 1*. $Ndp^{-/-}$ females are infertile, therefore the experimental cross to generate $Ndp^{KO};Ptch^{+/-}$ compound mutants and littermate controls must be performed by crossing $Ndp^{+/-}$ females with $Ptch^{+/-}$ males. Thus, the $Ndp^{KO}$ and $Ndp^{KO};Ptch^{+/-}$ genotypes are restricted to male mice carrying the $Ndp^{-/y}$ allele, whereas additional controls include both sexes. $Ptch^{+/-}$ (RRID:MGI:2177702), $Tie2$-$Cre$ (RRID:IMSR_JAX:008863), $Atoh1Cre$ (RRID:IMSR_JAX:011104), and $Fzd4^{fl/fl}$ (RRID:MGI:

4412187), mice were obtained from Jackson Laboratories and maintained on a C57BL/6 background. *Tie2Cre+;Fzd4^{fl/fl}* females are infertile, therefore *Tie2Cre+;Fzd4^{fl/fl};Ptch^{+/−}* compound mutants were generated by crossing *Fzd4^{fl/fl};Ptch^{+/−}* females with *Tie2Cre+;Fzd4^{fl/fl}* males. *Neurod2-Smo^{A1}* mice (*Hallahan et al., 2004*) were maintained as homozygotes (RRID:MGI:3831004), crossed with *Ndp^{+/−}* females, and tumors were monitored in *Neurod2-Smo^{A1+/−};Ndp^{−/y}* and *Neurod2-Smo^{A1+/−};Ndp^{+/y}* littermates. In every experiment, all compound mutants were compared to single mutant or wild-type controls from the same breeding cohort to ensure matched backgrounds. In Kaplan-Meier survival curve studies, mice were continually monitored and sacrificed upon display of advanced tumor symptoms or other adverse health effects as per CCAC endpoint guidelines. All animals were dissected to confirm presence or absence of medulloblastoma.

## Antibodies

The following primary antibodies were used for immunostaining:

| Antibody | Source | dilution |
|---|---|---|
| rabbit anti-mouse collagen IV | AbD Serotec 2150–1470 RRID:AB_2082660 | 1:1000 |
| rat anti-mouse CD31 | clone MEC13.3, BD Biosciences 550274 RRID:AB_393571 | 1:200 |
| rabbit anti-pan-laminin | Abcam ab11575 RRID:AB_298179 | 1:500 |
| rabbit anti-NeuN | Millipore ABN78 RRID:AB_10807945 | 1:1000 |
| rabbit anti-phospho-histone H3 | Millipore 06–570 RRID:AB_310177 | 1:500 |
| rat anti-mouse PLVAP | clone MECA-32, BD Biosciences 553849 RRID:AB_395086 | 1:100 |
| rabbit anti-mouse Claudin-5 | Thermo Fisher 34–1600 RRID:AB_2533157 | 1:500 |
| rat anti-mouse CD45R | clone RA3-6B2, BD Biosciences 550286 RRID:AB_393581 | 1:100 |
| rabbit anti-active caspase-3 | clone C92-605, BD Biosciences 559565 RRID:AB_397274 | 1:200 |
| rabbit anti-Pax6 (*Figure 1*) | Covance PRB-278P RRID:AB_2313780 | 1:500 |
| rabbit anti-Pax6 (*Figure 8—figure supplement 1*) | Thermo Fisher 426600 RRID:AB_2533534 | 1:300 |
| rat anti-myelin basic protein | AbD Serotec MCA409S RRID:AB_325004 | 1:100 |
| mouse anti-β-tubulin isotype III | clone SDL.3D10, Sigma-Aldrich T8660 RRID:AB_477590 | 1:1000 |
| rabbit anti-GFAP | Sigma-Aldrich G9269 RRID:AB_477035 | 1:1000 |
| rabbit anti-PDGF receptor beta | Abcam ab32570 RRID:AB_777165 | 1:100 |
| rat anti-mouse Endomucin | Santa Cruz sc-65495 RRID:AB_2100037 | 1:250 |
| mouse anti-ZO-1 | Thermo Fisher 339100 RRID:AB_2533147 | 1:300 |
| Rabbit anti-LEF1 | Clone C12A5, Cell Signaling 2230 RRID:AB_823558 | 1:250 |

## Tissue processing

For fixed tissue, pups younger than 14 days old were decapitated and brains were removed and placed directly into fixative. Animals 14 days and older were anesthetized and cardiac perfusion was performed using 10 ml of PBS then 10 ml of fixative, followed by dissection of the brain. Brains used for histological stains, Evans Blue visualization, in situ hybridization or immunostaining were then fixed in 4% paraformaldehyde (PFA) overnight at 4°C, whereas brains used for X-gal staining were fixed in 2% PFA with 2 mM MgCl$_2$ and 1.25 mM EGTA (ethylene glycol tetraacetic acid) for 45 min at 4°C. All tissues were then washed in PBS, cryoprotected in 30% sucrose/PBS overnight at 4°C, and embedded in a 50:50 mixture of OCT:30% sucrose by freezing in chilled 2-methylbutane. For fresh frozen tissue, unfixed brains were dissected and embedded as described above. If used for immunostaining, cardiac perfusion was performed using 10 ml of cold PBS before dissection. For laser capture microdissection, tissue was immediately dissected and embedded without perfusion, for maximum RNA integrity.

## Granule neuron progenitor (GNP) isolation

GNPs from the EGL or lesion-associated GNPs from P14 and P30 $Ptch^{+/-}$ mice were purified from cerebella by percoll gradient separation and pre-plating, as described previously (*Oliver et al., 2005*). Cells were then transferred to PDL-coated glass coverslips to proceed with immunostaining, or pelleted and immediately resuspended in lysis buffer for subsequent RNA extraction.

## Immunostaining

### PFA-fixed brain tissue

(immunostains in *Figure 1*, *Figure 3* top row, *Figures 7* and *8*) Brains were sectioned sagittally in a Leica CM1850 cyrostat at 12 µm onto Superfrost Plus positively charged slides (Fisher Scientific), air dried for 1–2 hr, and stored with desiccant at −20°C. Prior to immunostaining, antigen retrieval was performed in 10 mM sodium citrate buffer (pH 6) in a rice steamer. Slides were blocked with 10% normal serum in TBLS (tris-buffered saline containing 1% bovine serum albumin and 10 mM lysine) for 30 min at room temperature, and antibodies were diluted in TBLS and incubated overnight at 4°C. Sections were incubated with Alexa fluor secondary antibodies (Molecular Probes) at 1:1000 for 1 hr at room temperature, and nuclei were stained with Hoescht before coverslipping with fluorescence mounting medium (Dako S3023). For co-immunostains, both primaries and both secondaries were incubated simultaneously.

The following modifications were performed for peroxidase-based immunohistochemistry (IHC): Before blocking, endogenous peroxidases were quenched by incubating slides in 0.3% hydrogen peroxide in PBS for 15 min at room temperature. Biotinylated secondary antibodies were used at 1:200 (Dako), followed by incubation with avidin-biotin peroxidase (Vector Laboratories, PK-4000) for 40 min at room temperature, color development with 2.5% diaminobenzidine, and a brief counterstain with hematoxylin before mounting in 50:50 glycerol:PBS.

### Acetone-fixed brain sections

(immunostains in *Figure 3B*, *Figure 5*, *Figure 6*, *Figure 3—figure supplement 1*, *Figure 5—figure supplement 1* and *2*, *Figure 6—figure supplement 1* and *2*, *Figure 7—figure supplement 2*, and *Figure 8—figure supplement 1*) Fresh frozen brains were cryosectioned at 12 µm onto Superfrost Plus positively charged slides (Fisher Scientific), air dried for 1–2 hr, and slides were dipped in acetone for 20 s prior to storage at −80 until staining. Before staining, slides were fixed in ice cold acetone for 10 min, washed in ice cold 70% ethanol for 5 min followed by several washes in room temperature PBS, and blocked with 10% normal serum in PBS. Primary antibodies were diluted in 3% normal serum and incubated overnight at 4°C, followed by secondary detection and mounting as described above. For the Pax6/CD31/EdU triple stain, (*Figure 8—figure supplement 1*), the EdU was detected immediately following Pax6/CD31 immunostaining, using the Molecular Probes Click-iT EdU Alexa Fluor 647 Imaging Kit according to manufacturer's instructions.

### GNP immunostaining (*Figure 2*)

Once purified, GNPs were allowed to sit on PDL-coated glass coverslips for 2 hr in culture media, followed by the addition of 10 mM of anti-Fzd4 or anti-KLH antibodies to the media for 15 min, washing in PBS, fixation in 4% PFA, and secondary detection and mounting as described above.

### Retinal whole-mounts (*Figure 7—figure supplement 1*)

Eyes were removed and fixed in 2% PFA for 5 min. Retinae were dissected in 2X PBS, flattened by radial incisions, and stored in −20°C methanol until staining. Retinal whole-mounts (n = 3 biological replicates from each treatment, αFzd4 or αKLH) were blocked for 1 hr with 10% serum, 0.5% Triton-X, and 0.5% Tween-20 in PBS, incubated with Isolectin GS-IB$_4$ Alexa Fluor 594 Conjugate (1:100, Life Technologies I21413), transferred to slides, and coverslipped with Dako fluorescence mounting medium.

### X-gal staining (*Figure 1*)

Slides with 12 µm cryosections of 2% PFA-fixed brains were air-dried for 1 hr, placed in X-gal reaction buffer (1 mg/ml X-gal with 5 mM potassium ferrocyanide, 5 mM potassium ferricyanide, 2 mM

MgCl$_2$, 0.02% IGEPAL, 0.01% sodium deoxycholate in 0.1 M phosphate buffer), incubated overnight at 37°C, then washed and mounted with 50:50 glycerol:PBS. For co-X-gal/IHC stains, IHC was performed as described above, immediately after incubation in X-gal reaction buffer. For each X-gal stain or X-gal/IHC co-stain, at least 4 separate sections from n = 3 cerebella were examined.

### In situ hybridization (*Figure 3*)

Digoxigenin (DIG)-labeled antisense RNA riboprobes were prepared by in vitrotranscription from linearized plasmids containing complete or partial cDNA sequences of the following mouse genes: *Atoh1* (a gift from Dr. Carol Schuurmans), *Mycn*, and *Ccnd1*. ISH was performed as previously described (*Jensen and Wallace, 1997*), with the following modifications: Slides were incubated 1 to 5 hr in staining buffer containing NBT and BCIP, and slides were mounted in 50:50 glycerol:PBS. For each probe, at least 3 separate sections from n = 3 tumors were examined.

### Image acquisition

Brightfield images were visualized using an Axioplan microscope and captured with an Axiocam HRc camera. Fluorescent images were visualized using an AxioImager M1 microscope and captured with an Axiocam HRm camera, or a Zeiss AxioImager M2 microscope and Axiocam MRm camera. For the analysis of CD31/PH3 expression in *Figure 5*, CD31/Lef1 expression in *Figure 6—figure supplement 2*, and Pax6/CD31/EdU expression in *Figure 8—figure supplement 1*, images were captured using an LSM 780 confocal. Retinal whole-mount images were captured using an LSM 510 confocal. (All equipment from Carl Zeiss Inc.). Whole brain images of were captured using a Canon PowerShot SD1400 IS digital camera. Images were processed using Photoshop CS6 (Adobe).

### Fzd4 blocking antibody, pertussis toxin (Ptx) and EdU administration

Anti-Fzd4 and anti-KLH control antibodies were generated as described (*Paes et al., 2011*), and functionally tested in vivo before use (*Figure 7—figure supplement 1*). Antibodies were diluted in PBS to 3 mg/ml immediately prior to administration, and injected intraperitoneally at a dose of 30 mg/kg. *Ptch*$^{+/-}$ animals in lesion studies were injected once at P7, whereas *Ptch*$^{+/-}$ animals in the Kaplan-Meier survival study were injected at P7 followed by a booster dose at P16, an age when the phenotypic effects of a P7 injection are still present (*Paes et al., 2011*). Ptx was diluted in PBS to 5 mg/ml and pups received intraperitoneal injections of 120 ng Ptx or PBS vehicle control at P7, P9, P11 and P13. Animals were injected intraperitoneally with a dose of 10 mg/kg EdU 4 hr prior to sacrifice.

### EGL/lesion measurements and analysis of immunostains

For EGL/lesion measurements, sagittal 12 µm serial sections of cerebella were collected and examined along the mediolateral axis at intervals 144 µm apart, by hematoxylin and eosin (H and E) or cresyl violet staining, and blinded images at 5x magnification were captured. To quantify EGL thickness, three images (at identical locations at the cerebellar vermis between lobules VII and VIII) from n = 3 animals per genotype were measured. To quantify lesion number or volume, lesions were carefully followed continuously along the entire mediolateral axis, and scored as an individual lesion only if they remained spatially separate from all other lesions in every section. Lesion volume (mm$^3$) was calculated by measuring the 2-D area (mm$^2$) of each lesion section using ImageJ, multiplying it by the thickness separating each section from its neighbor (0.144 mm) to obtain the volume of each slice, and adding the individual slice volumes to obtain a total volume. To assess the vascularized, leaky vessel or CD45 accumulation status of the lesions, sagittal sections between 100 and 250 µm apart were examined along the entire mediolateral axis for Evans blue accumulation or stained with markers (anti-ColIV, laminin, CD31, PLVAP or CD45) to sample the entire lesion. To quantify lesions for PH3, cleaved caspase-3, NeuN, CD31, laminin or PLVAP immunostaining, or to examine PDGFRβ immunostaining, 3 to 4 blinded sections from each lesion at 10x or 20x magnification were analyzed. Using ImageJ, the number of PH3+ cells per unit area or percent area stained for caspase-3, NeuN, CD31, laminin or PLVAP were determined. Quantification involving lesion vasculature included all lesion-associated vessels, whether they were 1) on the outer surface of the cerebellum, 2) deeper into the cerebellar folds but still meningeal, or 3) growing into the 'lesion proper.' To quantify the number of PH3+ or Lef1+ cells per endothelial area, 3 sections per lesion were first imaged at 20x

magnification by epifluorescence to measure endothelial cell area via tracing of CD31+ vessels, and identify potential double labelled candidates. Sections were then examined by confocal microscopy to confirm double-labelled cells.

## Evans Blue injections and visualization

A 2% wt per volume Evans Blue (Sigma) solution in 0.9% saline was administered by intraperitoneal injection at 4 µl per gram of body weight and allowed to circulate overnight. Following perfusion and fixation in 4% PFA as described above, whole brains were photographed and 12 µm cryosections were then visualized under the far red fluorescence filter. The same or immediately adjacent section was H and E- or anti-ColIV-stained to provide a matched image.

## Laser capture microdissection

Fresh frozen cerebella were sectioned at 10 µm onto Superfrost microscope slides (Fisher Scientific) and placed immediately on dry ice before storage at −80°C for no more than 5 days before micro-dissection. Sections were stained and dehydrated by passing through RNase-free coplin jars with solutions made in DEPC-treated distilled water, as follows: 30 s in 75% ethanol, 30 s in distilled water, 2 min in 1% toluidine blue in distilled water, 30 s in distilled water, 30 s in 75% ethanol, 30 s in 95% ethanol, 1 min in 100% ethanol (2 times), and 5 min in xylene. All staining solutions except xylene contained RNase inhibitor (Sigma R7397). Slides were immediately microdissected using the Arcturus$^{XT}$ laser capture microdissection system (Life Technologies) in infrared mode, according to the manufacturer's instructions. Six to 10 sections from each lesion were captured onto CapSure macro LCM caps (Life Technologies), transferred immediately to RLT plus lysis buffer (Qiagen) with β-mercaptoethanol, briefly vortexed, and stored on dry ice until RNA extraction.

## RNA purification and quantitative RT-PCR

For microdissected lesion tissue, total RNA was extracted using the RNeasy Plus Micro kit (Qiagen) with genomic DNA eliminator columns, and amplified complementary DNA (cDNA) was prepared with the Ovation Pico WTA System V2 (NuGEN) according to the manufacturer's instructions. RNA integrity was assessed by an Agilent 2100 Bioanalyzer (Agilent Technologies) from slide scrapes. For all other samples, total RNA was extracted from freshly isolated/dissected GNPs, cerebellar tumor (with careful preservation of clean margins) or retina tissue using the RNeasy Mini Kit (Qiagen). First-strand cDNA was synthesized using the QuantiTect Reverse Transcription Kit (Qiagen), with and without reverse transcriptase to assess genomic contamination during downstream RT-PCR. For all samples, target gene mRNA levels were determined by quantitative RT-PCR (qRT-PCR) using iQ SYBR Green Supermix (Bio-Rad) and a MyiQ iCycler (Bio-Rad). Primer pairs were designed using PRIMER-blast (http://www.ncbi.nlm.nih.gov/), and are as follows:

*Gapdh* F: GGCCGGTGCTGAGTATGTCG, *Gapdh* R: TTCAAGTGGGCCCCGGCCTT, *Ndp* F: CCCACTGTACAAATGTAGCTCAA, *Ndp* R: AGGACACCAAGGGCTCAGA, *Fzd4* F: GACAAC TTTCACGCCGCTCATC, *Fzd4* R: CCAGGCAAACCCAAATTCTCTCAG, *Lrp5* F: GAGGAGTTC TCAGCCCATCC, *Lrp5* R: GATCAGGGGAGCAGGTAGGA, *Tspan12* F: GATTGCTGTCTGCTGC TTCC, *Tspan12* R: ACTGTACTGGCACCATAACCTC, *Ptch1 exons2-3* F: CTCCTCATATTTGGGGCC TT, *Ptch1 exons2-3* R: AATTCTCGACTCACTCGTCCA, *Gli1* F: ACATGCTGGTGGTGCACAT, *Gli1* R: AGGCGTGAATAGGACTTCCG, *Mycn* F: GCGGTAACCACTTTCACGAT, *Mycn* R: AGTTGTGCTGC TGATGGATG, *Esm1* F: ACAGGGTGACCGGAAGATGT, *Esm1* R: AGTCACGCTCTGTGTGGGAG, *Plvap* F: TGACTACGCGACGTGAGATG, *Plvap* R: CTCGCTCAGGATGATAGCGG, *Emcn* F: CTCCCGAAGGAACGACCAAAA, *Emcn* R: GGACCTTCAGTTGTTGTTCCC *Pecam1* F: GGAATACCAGTGCAGAGCGG, *Pecam1* R: CCTCGTTACTCGACAGGATGG, *Angpt2* F: AGAGGAGATCAAGGCCTACTGT, *Angpt2* R: GCCATCTTCTCGGTGTTGGA.

Primers were optimized using a 5-point standard curve of 2-fold diluted composite cDNA from relevant tissue and deemed acceptable with an $R^2 > 0.95$, a percent efficiency between 90-110%, a sharp single point melt curve, positive controls with Ct values > 10 cycle difference compared to no RT control samples, and expected amplicon size by agarose gel electrophoresis. All samples were run in triplicate, normalized to *Gapdh*, and quantified relative to the reference tissue indicated. During qRT-PCR for microdissected lesions, the expression of genes known to be highly expressed in lesions (*Gli1*, *Mycn*) were assessed in parallel to *Ptch*, along with microdissected tumor samples

known to have *Ptch* LOH, or microdissected EGL known to have high levels of *Ptch* expression. *Ptch* LOH status was assigned by quantification relative to microdissected tumors with known LOH.

## Microarray analysis of mouse tissue

Total RNA extracted from MB or GNPs was analyzed with the MouseWG-6 v2.0 Expression Bead-Chip array platform. Illumina BeadStudio outputs were analyzed and annotated with R packages limma (*Ritchie et al., 2015*) and illuminaMousev2.db version 1.26.0. Data were processed with neqc function (*Shi et al., 2010*). The hierarchical clustering and the principal component analysis plots were prepared from the 1500 most variable probes across all samples in terms of interquartile range, after data processing (background correction, normalization, and log-transformation), and filtering out probes annotated as Bad and No Match. Significant changing probes were detected with the linear modeling approach and empirical Bayes statistics of limma (*Smyth, 2004*). Gene Ontology (GO) cellular components enrichment was investigated using DAVID (*Huang et al., 2009*) for probes with adjusted $P$ value below 0.05 and higher fold changes (>1.0) between $Ndp^{KO};Ptch^{+/-}$ and $Ptch^{+/-}$ tumors. The tables in *Figure 3* display the GO terms with $P$ values<0.05. In an analogous manner we analyzed samples from purified P6 GNPs. The principal component analysis plot was prepared from the 1,500 most variable probes as above.

## Human tumor samples and expression analysis

*NDP* expression profiles were determined across three independent cohorts with the R2 database analysis tool (http://r2.amc.nl) using publically available datasets from Heidelberg (*Remke et al., 2011*), Toronto (*Northcott et al., 2011*) and Boston (*Cho et al., 2011*). The following probes were used for analysis: Toronto (Affymetrix Exon 1.0 T Probe Accession: 4006280), Heidelberg (Agilent 4 x 44 k Probe Accession: A_23_P73609) and Boston (Affymetrix u133a Probe Accession: 206022_at). $P$ values represent ANOVA across the four subgroups. Survival analysis in *Figure 1* was performed on the MAGIC dataset of clinically annotated SHH tumors (Affymetrix Human Gene 1.1 ST, Probe Accession: 8172220) (*Northcott et al., 2012*; *Vanner et al., 2014*). Log$_2$ transformed expression values of *NDP* were then ranked as either the bottom 10th percentile of expression versus the top 10th percentile. Survival was calculated using the log-rank method in the R-Statistical Environment (v3.1.3) using packages survival (v2.37–7) and ggplot2 (v1.0.0).

## Statistics

Sample sizes are as reported in figures and figure captions. Mice were randomly assigned to either αFzd4 or Ptx treatment groups. Statistics were determined by GraphPad Prism 6 or SPSS software. The logrank test was used to generate Kaplan Meier survival curves (*Figures 1*, *2*, *7* and *8*). Lesion number (*Figures 4* and *7*), vessel or laminin density (*Figures 5* and *6*), number of endothelial PH3+ cells (*Figure 5*), qRT-PCR (*Figures 1 –* and *3*) and EGL thickness (*Figure 4—figure supplement 1*) were analyzed by one-way ANOVA with Tukey *post-hoc* comparisons. Comparison of lesion volumes (*Figures 1*, *4* and *7*), immunostain quantification (*Figure 8*), and tumor latency (*Figures 2*, *7* and *8*) were analyzed by a two-tailed, unpaired Student's *t*-test. The proportion of vascularized, leaky or CD45+ lesions (*Figures 5* and *6*) and tumor incidence (*Figures 2*, *7* and *8*) were analyzed by the hypergeometric test. The number of endothelial Lef1+ nuclei (*Figure 6—figure supplement 2*) was assessed by a one-way ANOVA and Fischer's LSD *post hoc* comparison. A Levene's test for homogeneity of variance and normality tests were used to verify parameters for parametric analysis. Microarray analyses were analyzed as described above.

## Acknowledgements

For technical advice, we thank Dr. L Murray (laser capture microdissection), Drs. A Prat and JI Alvarez (immunostaining of acetone-fixed tissue), Dr. Phil Nickerson (data analysis), and Robin Vigouroux (Ptx administration). We thank Drs. R Bremner, F Charron, R Kothary, D Picketts and D van der Kooy for helpful discussions and manuscript comments. NT is a recipient of the Queen Elizabeth II Graduate Scholarship in Science and Technology. This work was supported by the Canadian Cancer Society (grant # 701740) and the Cancer Research Society (grant # 19188). The αFzd4 antibody

was obtained under a Materials Transfer Agreement between Lexicon Pharmaceuticals, Inc. and The Ottawa Hospital Research Institute.

## Additional information

### Funding

| Funder | Grant reference number | Author |
|---|---|---|
| Cancer Research Society | 19188 | Valerie A Wallace |
| Canadian Cancer Society Research Institute | 701740 | Valerie Wallace |

The funders had no role in study design, data collection and interpretation, or the decision to submit the work for publication.

### Author contributions

EAB, Conception and design, Performed experiments and interpreted and analyzed data, Designed and prepared figures, Drafted and revised the manuscript; NT, Conception and design, Performed experiments and interpreted and analyzed data, Assisted with figure preparation; EAA, NTP, Performed experiments, Analyzed data; CM, Provided technical assistance with mouse colony management and experiments; KM, Provided technical assistance with mouse colony management; AJM, Provided conceptual and technical advice regarding mouse gene expression experiments, Analyzed mouse microarray data; BM, Conception and design of experiments; RR, CC, Conception, design and technical advice and assistance; SS, Provided technical assistance and mouse colony management; AMD, Performed human tumor expression analyses; VR, MR, Performed human tumor expression and survival analyses; PAN, Acquired and analyzed human tumor expression data; PPM, Conception, design and technical advice, Critical reading of manuscript; DP, Produced the function blocking anti-Frizzled4 antibody; KP, Conception, design and technical advice, Designed and generated the function blocking anti-Frizzled4 antibody, Critical discussions and reading of manuscript; LLK, KJC, Generated the function blocking anti-Frizzled4 antibody; DSR, Conception, design and technical advice, Design of the function blocking anti-Frizzled4 antibody, Critical discussions and reading of manuscript; CP-I, Critical discussion and analysis of mouse microarray data, Assisted with figure, figure legend and methods preparation; MDT, Conception, design and technical advice, Assistance with human tumor data access and analysis, Critical discussions and reading of manuscript; VAW, Conception, design and supervision of all experiments, Data interpretation, Drafted and revised the manuscript, Critical discussions

### Author ORCIDs

Valerie A Wallace, http://orcid.org/0000-0003-3721-9017

### Ethics

Animal experimentation: All experiments were approved by the University of Ottawa and University Health Network Animal Care Ethics Committees and adhered to the guidelines of the Canadian Council on Animal Care.

## Additional files

### Major datasets

The following dataset was generated:

| Author(s) | Year | Dataset title | Dataset URL | Database, license, and accessibility information |
|---|---|---|---|---|
| Erin A Bassett, Nicholas Tokarew, Ema A Allemano, Chantal Mazerolle, Katy Morin, Alan J Mears, Brian McNeill, Randy Ringuette, Charles Campbell, Sheila Smiley, Neno T Pokrajac, Adrian M Dubuc, Vijay Ramaswamy, Paul A Northcott, Marc Remke, Philippe P Monnier, David Potter, Kim Paes, Laura L Kirkpatrick, Kenneth J Coker, Dennis S Rice, Carol Perez-Iratxeta, Michael D Taylor, Valerie A Wallace | 2016 | Norrin-dependent gene expression in murine cerebellar granule neuron progenitors and Patched medulloblastoma | http://www.ncbi.nlm.nih.gov/geo/query/acc.cgi?acc=GSE81964 | Publicly available at NCBI Gene Expression Omnibus (accession no: GSE81964) |

The following previously published datasets were used:

| Author(s) | Year | Dataset title | Dataset URL | Database, license, and accessibility information |
|---|---|---|---|---|
| Remke M, Hielscher T, Korshunov A, Northcott PA, Bender S, Kool M, Westermann F, Benner A, Cin H, Ryzhova M, Sturm D, Witt H, Haag D, Toedt G, Wittmann A, Schöttler A, von Bueren AO, von Deimling A, Rutkowski S, Scheurlen W, Kulozik AE, Taylor MD, Lichter P, Pfister SM | 2011 | Human Medulloblastoma Samples | http://www.ncbi.nlm.nih.gov/geo/query/acc.cgi?acc=GSE28245 | Publicly available at NCBI Gene Expression Omnibus (accession no: GSE28245) |
| Northcott PA, Korshunov A, Witt H, Hielscher T, Eberhart CG, Mack S, Bouffet E, Clifford SC, Hawkins CE, French P, Rutka JT, Pfister S, Taylor MD | 2012 | Subgroup specific somatic copy number aberrations in the medulloblastoma genome [mRNA] | http://www.ncbi.nlm.nih.gov/geo/query/acc.cgi?acc=GSE37382 | Publicly available at NCBI Gene Expression Omnibus (accession no: GSE37382) |
| Kool M, Koster J, Bunt J, Hasselt NE, Lakeman A, van Sluis P, Troost D, Meeteren NS, Caron HN, Cloos J, Mrsić A, Ylstra B, Grajkowska W, Hartmann W, Pietsch T, Ellison D, Clifford SC, Versteeg R | 2008 | mRNA expression data of 62 human medulloblastoma tumors | http://www.ncbi.nlm.nih.gov/geo/query/acc.cgi?acc=GSE10327 | Publicly available at NCBI Gene Expression Omnibus (accession no: GSE10327) |

| Northcott PA, Korshunov A, Witt H, Hielscher T, Eberhart CG, Mack S, Bouffet E, Clifford SC, Hawkins CE, French P, Rutka JT, Pfister S, Taylor MD | 2011 | Genomics of medulloblastoma identifies four distinct molecular variants | http://www.ncbi.nlm.nih.gov/geo/query/acc.cgi?acc=GSE21140 | Publicly available at NCBI Gene Expression Omnibus (accession no: GSE21140) |
| Vanner RJ, Remke M, Gallo M, Selvadurai HJ, Coutinho F, Lee L, Kushida M, Head R, Morrissy S, Zhu X, Aviv T, Voisin V, Clarke ID, Li Y, Mungall AJ, Moore RA, Ma Y, Jones SJ, Marra MA, Malkin D, Northcott PA, Kool M, Pfister SM, Bader G, Hochedlinger K, Korshunov A, Taylor MD, Dirks PB | 2014 | Sox2 signature in SHH medulloblastomas | http://www.ncbi.nlm.nih.gov/geo/query/acc.cgi?acc=GSE50765 | Publicly available at NCBI Gene Expression Omnibus (accession no: GSE50765) |

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
