## [Decision Letter]

Thank you for submitting your article "Norrin/Frizzled4 signalling in the preneoplastic niche blocks medulloblastoma initiation" for consideration by *eLife*.Your article has been reviewed by three peer reviewers, and the evaluation has been overseen by a Reviewing Editor and Fiona Watt as the Senior Editor. The reviewers have opted to remain anonymous.

The reviewers have discussed the reviews with one another and the Reviewing Editor has drafted this decision to help you prepare a revised submission. Although the required revisions enumerated below may take a bit longer to complete than the two months we suggest as a time limit for the return of a revised manuscript, the reviewers feel the effort will be worthwhile and could strengthen the conclusions you wish to draw. Please respond with your plans to address these issues and an estimate of the length of time you will require to complete the revisions.

Summary:

This is a very interesting paper that addresses an under studied area of medulloblastoma (Mb) research, namely the interactions between tumor cells and the stromal environment that influence tumor incidence and growth rate. The authors convincingly show that removing the secreted factor Ndp from mice, or the Fzd4 receptor specifically in endothelial cells, results in a shorter tumor latency and increased incidence in a SHH-Mb model (*Ptch1* heterozygous mice). NDP-FZD4 signaling thus appears to modify the endothelial cells in the microenvironment to be tumor protective. The demonstration of a tumor-suppressive function for NDP-FZD4 signaling in the vascular microenvironment of MB is novel, although the NDP-FZD4 signaling pathway has been shown by others to be required for maintaining the blood brain barrier in the vasculature surrounding the cerebellum. They also show that NDP expression is higher in human SHH versus other Mb subgroups and that low NDP RNA expression trends towards lower survival of patients with SHH-MBs. They propose that vascular remodeling stimulates the transition of pre-neoplastic tissue to malignancy. However, this is based almost entirely on analyzing small lesions in P14 mice, long before tumors arise and the vast majority of which will not go on to form a tumor. In addition, some of the quantification of the data and interpretation of the results does not fully support this conclusion as opposed to others. Although an ideal experiment would be to use a tumor model where NDP can be removed only in the tumor cells, the acute blocking of FZD4 and conditional deletion in endothelial cells provide support that the increased tumor incidence is not due to a developmental defect and relies on signaling to blood vessels. With additional data that support the role for vascularization, this paper will be an important contribution to the cancer field.

Essential revisions:

The expression pattern of Ndp-lacZ resembles HH target gene expression. If Ndp is a HH target gene, then this would mean that in *Ptch* heterozygotes the pathway is elevated in all cell types, and hyperactivated in tumor cells. This might explain why NDP is only highly expressed in the SHH subgroup of medulloblastoma. Given the increase in tumor incidence when Ndp is removed in mice and proposed resulting increase in vascularization, loss of BBB and increased LOH in early lesions raises the question of whether the SHH-subgroup tumors have more or less blood vessels than other subgroups and whether blood vessel content and BBB correlates with NDP expression levels in SHH-MB. If human MB sections/samples are not available, gene expression analysis of human tumors should be performed (as was done for the mouse microarray data) to determine whether blood vessel marker genes are differentially expressed in the SHH-subgroup, and whether NDP expression levels correlate with blood vessel and BBB markers.

Similarly, given that all MBs are vascularized, and the BBB was recently shown to be compromised only in the WNT subgroup (Phoenix, Cancer Cell, 2016) the expression data presented suggest that in the Group3 and Group4 Mb, BBB integrity is maintained by a signaling mechanism not involving NDP-FZD4. Can the Taylor and Northcott groups mine their human Mb expression data sets to analyze this further?

A more detailed analysis is needed of the cell types that express Ndp using double immune staining for LacZ and markers of GNPs (PAX6) and astrocytes or endothelial cells, and in early lesions and tumor sections (RNA in situs could be used). Also, the authors need to address whether GNPs and tumor cells express the NDP receptor.

The quantification of the blood vessels at P14 is a concern. Based on the images in Figure 1, there is a correlation between the amount of blood vessels and the size of the lesion. Is this generally the case? If so, the authors need to rethink their interpretation. Perhaps the size of the lesion determines whether new blood vessels (from the meninges or elsewhere) will invade the lesion. If the size of the lesion determines whether blood vessels invade, then another interpretation of the mutant studies is that Ndp/Fzd4 loss increases the likelihood of lesions forming and/or their growth rate and occurrence of LOH.

The expression level of Ndp in highly vascularized lesions should be compared to those that are not vascularized. If it is not decreased in the vascularized lesions and tumors, how do the authors explain the result?

Also, removal of the meninges from the quantification is problematic. In Figure 1, Figure 5 double mutants and 6D "angiogenic", some of the blood vessels could well be meningeal, but they are not running in a straight line because the lesions are large and likely distort or change the organization of meningeal blood vessels. If they are not included in the quantification in B, this will push the numbers away from the phenotype being claimed.

The characterization of angiogenic vessels should be substantiated. Collagen IV expression was used to indicate angiogenic endothelial cell (e.g. Figure 6). However, these types of vessels might be defined as remodeling vessels. Typically, to convincingly indicate an angiogenic endothelial cell, proliferation and other classical markers such as Notch ligands are needed.

Figure 3. Can quantification be added, especially for the vascular content? A shortcoming of the paper is that all the quantification is performed at P14, long before tumors form. RNA-seq data is presented to support that vascularization is increased in the tumors, but given the heterogeneity of tumors cells, this is problematic. For example, is there an increase in leukocytes in double mutant tumors and is vascular permeability different in tumors with and without NDP-FZD4 signaling (Evan's blue and PLVAP/CLD5 expression)? If large tumors are difficult to analyze, an earlier stage should be analyzed.

Also, to better address mechanism and the question of whether the neoplastic transition coincides with recruitment/infiltration of inflammatory cells, the authors should determine if there is a correlation between the location and density of inflammatory cells (CD45-positive leukocyte) and neoplastic transition at a later stage than P14 when tumors are forming and whether there is a difference when Ndp or Fzd4 are removed.

Figure 5 and Figure 6. The Fzd4 mutant lesions are very curious or interesting, as only the inside region has vessels. Is this seen consistently? One interpretation of 5A is that the meningeal blood vessel was split in two by premalignant cells and the vessel now surrounds the lesion. Also, are the GNPs surrounding the vessels proliferating or postmitotic and the ones invaded by blood vessels are they different?

The experiment with a single dose of αFzd4 administered as late as P7 showing that it was sufficient to induce the cellular changes seen in the conditional and null mutants within 7 days (by P14) is a very striking result. It is important to determine whether LOH is increased in these lesions, in order to test whether LOH is related to the phenotype.

PLVAP was upregulated in endothelial cells when NDP-FZD4 signaling was suppressed. PLVAP is known to be essential for formation of vascular fenestration. Does PLVAP upregulation increase fenestration in the mutant tumor vasculature?

---

## [Author Response]

*[…] Essential revisions:*

*The expression pattern of Ndp-lacZ resembles HH target gene expression. If Ndp is a HH target gene, then this would mean that in Ptch heterozygotes the pathway is elevated in all cell types, and hyperactivated in tumor cells. This might explain why NDP is only highly expressed in the SHH subgroup of medulloblastoma. Given the increase in tumor incidence when Ndp is removed in mice and proposed resulting increase in vascularization, loss of BBB and increased LOH in early lesions raises the question of whether the SHH-subgroup tumors have more or less blood vessels than other subgroups and whether blood vessel content and BBB correlates with NDP expression levels in SHH-MB. If human MB sections/samples are not available, gene expression analysis of human tumors should be performed (as was done for the mouse microarray data) to determine whether blood vessel marker genes are differentially expressed in the SHH-subgroup, and whether NDP expression levels correlate with blood vessel and BBB markers.*

*Similarly, given that all MBs are vascularized, and the BBB was recently shown to be compromised only in the WNT subgroup (Phoenix, Cancer Cell, 2016) the expression data presented suggest that in the Group3 and Group4 Mb, BBB integrity is maintained by a signaling mechanism not involving NDP-FZD4. Can the Taylor and Northcott groups mine their human Mb expression data sets to analyze this further?*

The relationship between tumor group and blood vessel density has been addressed recently by Phoenix et al. (2016), who used quantitative IHC on sectioned tumor tissue and showed that the vascular density of human Wnt MBs was 4 times greater than that of human Shh, Group 3 or Group 4 MBs and that Wnt, but not Shh tumors, have disruption of the BBB. These tumor subtype-dependent vascular phenotypes were associated with altered expression of secreted Wnt inhibitors, with higher levels reported in Wnt MB. Our study extends this concept to implicate Ndp expression as another factor that contributes to BBB integrity in mouse and human Shh MB.

The reviewers raise an excellent point regarding the impact of Ndp levels on vascularity and BBB integrity within the human Shh MB group. Because is not feasible to compare Ndp high and low tumor material by IHC, we addressed whether there were differences in gene expression between NDP^low^ and NDP^high^ Shh MB, albeit looking at bulk tumor, from which the overwhelming majority of signal will be from tumor cells rather than from the vasculature. This analysis revealed few (6) differentially expressed genes, only one significantly enriched geneset (phosphoinositide binding) and no significant enrichment for a specifically curated angiogenesis geneset.

That we did not detect significant changes in angiogenic or BBB genes could be due to the small sample size of Ndp^low^ tumors and the low frequency of endothelial cells within tumors. Consistent with the latter point, we note that in the Phoenix et al. (2016) study, the differences in vascularity/BBB between MB subtypes was not identified on the basis of gene expression analysis from whole tumor transcriptome data.

Although the reviewer makes interesting suggestions, as all of the published genomic data was generated using bulk tumor, our ability to mine for cell lineage/signaling pathways that could regulate BBB integrity in Group 3 and 4 tumors, is likely not feasible and is certainly a multi-year undertaking that would be more appropriate for future studies.

*A more detailed analysis is needed of the cell types that express Ndp using double immune staining for LacZ and markers of GNPs (PAX6) and astrocytes or endothelial cells, and in early lesions and tumor sections (RNA in situs could be used). Also, the authors need to address whether GNPs and tumor cells express the NDP receptor.*

[…]

*The expression level of Ndp in highly vascularized lesions should be compared to those that are not vascularized. If it is not decreased in the vascularized lesions and tumors, how do the authors explain the result?*

Additional (paraphrased) comments from editor:

How reliable is the Ndp-LacZ as a reporter for Ndp expression?Were males used for the Ndp-LacZ reporter analysis?Why were the RT-PCR data for Ndp expression normalized to Ndp KO samples?The expression of NDP in human medulloblastoma is the same as in adult cerebellum, so that would indicate it is quite low.Given that the Ndp KO allele is the lacZ reporter allele, why have the authors not analyzed lacZ expression in the early lesions or medulloblastoma samples in males mutant for Ndp (on the X chromosome) or the control female littermates they should have that are heterozygous for the KI allele and Ptch1^+/-^?At this point, the editor thinks that the author's idea of analyzing Lef1 expression (a canonical Wnt target) in vascularized and non-vascularized Ptch^+/-^ lesions is a good one, but would like to hear a response to the comments above and see some definitive proof added to the paper that Ndp is expressed in GNPs and early lesions.

1) Provide evidence that Ndp and signaling pathway components are expressed in GNPs.

We show that β-gal activity from the Ndp-LacZ reporter overlaps with expression of proliferation (Ph3) and GNP (Pax6) markers in the EGL of the developing cerebellum (Figure 1). The reviewer raises the question of whether the design of the Ndp-LacZ knockin allele results in a faithful reporter for this gene. While the presence of the targeting cassette can interfere with transcription of the locus, it would result in a reduction of reporter gene expression rather than ectopic expression. Thus, even if the Ndp-LacZ allele is under reporting the expression of this gene, we are still able to detect reporter expression in the EGL. We also clarify that male Ndp^-/Y^ mice were used for the reporter gene analyses.

We agree that it would be desirable to corroborate the Ndp-LacZ reporter expression with RNA in situ hybridization or IHC for Ndp with lineage markers. Despite several attempts, this approach has not been successful. Because of these limitations we used q-RT-PCR to confirm the presence of transcripts for Ndp in purified GNPs and established MB (Figure 1). In the original submission we normalized the NDP expression levels against NDP KO samples because the primers we used do not amplify the null allele. Because representing the data in this manner is unconventional (and confusing), we now report the Ndp levels relative P7 GNPs (after normalizing to Gapdh) (revision, Figure 1). While transcript levels are reduced in MB relative to P7 GNPs, they are still detectable (average Ct values: P7 GNPs, 24.0 Ndp, 17.3 Gapdh n= 4; MB, 26.5 Ndp, 19.0 Gapdh, n=4). Moreover, our expression data is corroborated in published gene expression dataset where Ndp expression is detected in GNPs and MB (GSE2426 (Oliver et al., 2005);GSE34126 (Pei et al., 2012)).

2) Do GNPs and tumor cells express the NDP receptor?

We used q-RT-PCR to detect transcripts for Fzd4, Lrp5 and TSPAN12 in purified GNPs and MB samples (Figure 2). We also now provide new data showing IHC for Fzd4 in purified GNPs (revision, Figure 2).

In summary, we are confident in concluding that GNPs and MB express Ndp and components of the NDP signal transduction pathway.

3) What cell types in the cerebellum express NDP?

The identity of the other Ndp-expressing cells in the cerebellum is interesting, but not easily resolved given the technical challenges we describe above. However, Phoenix et al. (2016) reported Ndp expression in endothelial cells isolated from mShh and mWnt MB. Moreover, the pattern of reporter activity in an Ndp-AP reporter mouse line is consistent with expression of this gene in Bergman glia (BG) (Ye et al., 2011). Unfortunately, the distribution of the AP reporter activity in BG obscures signal in the EGL (the BG fibers extend into the EGL), therefore one cannot draw conclusions about expression in the GNPs residing in the EGL is the Ndp-AP reporter line. While the cellular source of Ndp that mediates BBB stability is an important question, addressing it would require cell type-specific inactivation of Ndp in GNPs and BG, which it is beyond the scope of the current study.

4) What is the relationship between Ndp expression and vascular state of lesions in *Ptch* mice and if Ndp expression is not changed then how do we account for the vascularization of lesions in *Ptch^+/-^* mice.

The reviewer raises an excellent point regarding the relationship between Ndp expression and lesion vascularization in *Ptch^+/-^* mice. However, in light of the technical challenges of detecting Ndp expression (outlined above) and because we cannot use X-gal staining from the Ndp-LacZ reporter in the *Ptch^+/-^* background (the mutant *Ptch* allele is also a lacZ knock in), it is not feasible to use this approach to monitor Ndp expression in lesions in situ. Therefore, we have used two complimentary approaches to address this issue. First, we compared Ndp expression levels by q-RT-PCR in wildtype GNPs and *Ptch^+/-^* GNPs isolated from the cerebellum along a continuum of tumorigenic stages. Here we find that Ndp expression is reduced as a function of tumor progression, with lower levels of Ndp detected in lesion stage GNPs at P30 and MB relative to P7-P14 GNPs (revision Figure 1). Second, we compared endothelial cell expression of Lef1, a canonical Wnt reporter marker, in vascularized and non-vascularized lesions in *Ptch^+/-^* mice. The rationale for this approach is that it would be a general reporter of perturbed canonical Wnt signaling in endothelial cells. Here we found that there was a significant reduction in Lef1 expression in endothelial cells in vascularized *Ptch^+/-^* lesions relative to EGL-associated vessels from non-lesion regions (revision, Figure 6—figure supplement 2). While we do not think that the reduction in Lef1 reflects a complete inactivation of endogenous Ndp/Fzd4 signaling in *Ptch^+/-^* lesions (because we do not observe the same disruption of the BBB that we do in Ndp or Fzd4 mutants), our findings do suggest the possibility that downregulation of endogenous canonical Wnt signaling in endothelium is associated with lesion vascularization. We speculate that reduced Wnt signaling in endothelial cells sensitizes EGL-associated vessels to respond to angiogenic signals that normally drive vascularization in these lesions/tumors. Genetic disruption of Ndp/Fzd4 signaling, therefore, represents a more strongly sensitized model for this process.

These experiments link our mutant mouse analysis to endogenous changes in *Ptch^+/-^* tumor progression and strengthen our manuscript. We thank the reviewers for these suggestions.

5) The authors state "*Ndp-^KO^;Ptch^+/-^* compound mutants must be generated by crossing *Ndp^+/-^* females with *Ptch^+/-^* males, resulting in all male (*Ndp^-/y^;Ptch^+/-^*) compound mutants", which does notmake sense since half the males would receive a wild type Ndp allele from the female.

We agree and have corrected this mis-statement in the Methods section.

*The quantification of the blood vessels at P14 is a concern. Based on the images in Figure 1, there is a correlation between the amount of blood vessels and the size of the lesion. Is this generally the case? If so, the authors need to rethink their interpretation. Perhaps the size of the lesion determines whether new blood vessels (from the meninges or elsewhere) will invade the lesion. If the size of the lesion determines whether blood vessels invade, then another interpretation of the mutant studies is that Ndp/Fzd4 loss increases the likelihood of lesions forming and/or their growth rate and occurrence of LOH.*

We agree with the reviewer that the relationship between lesion size and vascular density is an important factor to consider. Indeed we investigated this possibility at the outset of our study, based on our data showing that lesions in the compound mutants exhibited features of more advanced tumors at early stages. It is for this reason that we compared lesion volume at P14 our double mutant models. While the vast majority of lesions are vascularized in the double mutant models (78% *Ndp^KO^;Ptch^+/-^* and 95% *Tie2Cre;Fzd4^fl/fl^;Ptch^+/-^* vs. 24% *Ptch^+/-^*) we did not observe a change in lesion volume (Figure 4). Therefore we would argue that vascular density and lesion volume are independent in our double mutant models, at least at this early stage in tumor evolution. To further illustrate this point, we provide examples of vascular remodeling in very small double mutant lesions (revision, Figure 5—figure supplement 2).

However, in single *Ptch^+/-^* mice we do find a significant difference in lesion volume in vascularized compared with non-vascularized lesions at P14 (revision, Figure 1), which is consistent with the possibility that activation of angiogenesis is a progression event in single *Ptch^+/-^* mice. This relationship is also consistent with the accelerated tumor progression we observe in *Ndp^KO^;Ptch^+/-^ Tie2Cre;Fzd4^fl/fl^;Ptch^+/-^* mice. However, EGL proliferation (prior to lesion formation) and lesion volume are not increased in the double mutants, which argues against a simple model of vessel invasion triggered by increased growth of GNPs. We do favor the second interpretation of our findings, e.g. that altered Ndp/Fzd4 signaling in endothelial cells promotes lesion formation and an imbalance of proliferation/death resulting in a net gain of proliferation and greater opportunity for *Ptch* LOH and hence progression.

*Also, removal of the meninges from the quantification is problematic. In Figure 1, Figure 5 double mutants and 6D "angiogenic", some of the blood vessels could well be meningeal, but they are not running in a straight line because the lesions are large and likely distort or change the organization of meningeal blood vessels. If they are not included in the quantification in B, this will push the numbers away from the phenotype being claimed.*

We would like to clarify that we did include meningeal vessels in the quantification. Essentially all vessels in lesions whether they were 1) on the surface, 2) deeper into the folds but still meningeal, or 3) growing into the "lesion proper" were included in the quantifications. We clarify this point in the Methods section. Thus, we do not disagree that some of the vessels included in the quantification of vascular density (Figure 5) or angiogenic score (Figure 1, Figure 5, Figure 7) could be displaced meningeal vessels, as opposed to newly generated vessels via angiogenesis. Moreover, even if they are displaced meningeal vessels, they do contribute to overall vessel density. We take the reviewers’ point that describing lesions as “angiogenic” is confusing, since the assignment is not based on analysis of bona fide angiogenic markers. Therefore, we revised our terminology in these figures to describe the lesions as vascularized vs. non-vascularized with the latter representing lesions where vessels are restricted to the surface.

*The characterization of angiogenic vessels should be substantiated. Collagen IV expression was used to indicate angiogenic endothelial cell (e.g. Figure 6). However, these types of vessels might be defined as remodeling vessels. Typically, to convincingly indicate an angiogenic endothelial cell, proliferation and other classical markers such as Notch ligands are needed.*

To address this concern we quantified mitotic (Ph3+) endothelial cells in *Ptch^+/-^* and *Ndp^KO^;Ptch^+/-^* lesions at P14 and show that the frequency of these cells as a function of vessel area is significantly increased in the *Ndp^KO^;Ptch^+/-^* lesions (revision, Figure 5).

*Figure 3. Can quantification be added, especially for the vascular content? A shortcoming of the paper is that all the quantification is performed at P14, long before tumors form. RNA-seq data is presented to support that vascularization is increased in the tumors, but given the heterogeneity of tumors cells, this is problematic. For example, is there an increase in leukocytes in double mutant tumors and is vascular permeability different in tumors with and without NDP-FZD4 signaling (Evan's blue and PLVAP/CLD5 expression)? If large tumors are difficult to analyze, an earlier stage should be analyzed.*

The reviewer raises an excellent point regarding issue of quantifying phenotypic markers in established tumors. Because of sampling bias and intra-tumoral heterogeneity it is difficult to draw conclusions regarding marker and gene expression. For the lesion analyses, all of the quantification was performed on step sections through entire lesions. However, this approach is not practical for large tumors. Therefore, we used global approaches (microarray and q-RT-PCR) and showed upregulated expression of vascular, angiogenic and BBB genes in *Ndp^KO^;Ptch^+/-^* compared with *Ptch^+/-^* MB (Figure 3). In addition, we also corroborated increased PLVAP expression in *Ndp^KO^;Ptch^+/-^* tumors by IHC (Figure 3—figure supplement 1). It is notable that we detected an increase in PLVAP mRNA in whole *Ndp^KO^;Ptch^+/-^* MB samples, particularly since the contribution of endothelial cells relative to tumor cells is relatively small. To further illustrate the point that BBB disruption is maintained from lesion stage to MB, we now include images of whole mount Evan’s Blue staining of tumors from our mouse models (revision, Figure 3). Taken together these data confirm that the BBB disruption and increased vascularity that we document in early lesions are maintained in established tumors with deficient Ndp/Fzd4 signaling. How these and other features, including immune infiltration, contribute to tumor growth is an excellent avenue for future investigation.

*Also, to better address mechanism and the question of whether the neoplastic transition coincides with recruitment/infiltration of inflammatory cells, the authors should determine if there is a correlation between the location and density of inflammatory cells (CD45-positive leukocyte) and neoplastic transition at a later stage than P14 when tumors are forming and whether there is a difference when Ndp or Fzd4 are removed.*

We agree with the reviewer that the relationship between inflammation and tumor progression in our model is an interesting direction to explore. We consistently observe infiltration of CD45+ cells in lesions of double mutant, but rarely single *Ptch^+/-^* mice (revision, Figure 6). Investigating the role of inflammation as a function of progression is a very involved undertaking that would require longer than the revision period to complete. I would trust that the reviewer would agree that this line of investigation is more appropriate for future studies.

*Figure 5 and Figure 6. The Fzd4 mutant lesions are very curious or interesting, as only the inside region has vessels. Is this seen consistently? One interpretation of 5A is that the meningeal blood vessel was split in two by premalignant cells and the vessel now surrounds the lesion. Also, are the GNPs surrounding the vessels proliferating or postmitotic and the ones invaded by blood vessels are they different?*

We thank the reviewer for this observation. In early lesions at P14 it is very likely that the meningeal vessels, which are already present in the area of the newly forming lesion and, notably, are most dependent on Ndp/Fzd4 signaling for BBB integrity, are involved in vascular remodeling. Therefore, it will typically look like there are more vessels in the centre of the early lesions. We provide images showing the relationship between Pax6+EdU+ (dividing GNPs) and blood vessels in lesions from *Ptch* and *Ndp^KO^;Ptch^+/-^* mice (revision, Figure 8—figure supplement 1). These data show that Pax6+ GNPs are located in the vicinity of blood vessels and are in S-phase (EdU+).

*The experiment with a single dose of αFzd4 administered as late as P7 showing that it was sufficient to induce the cellular changes seen in the conditional and null mutants within 7 days (by P14) is a very striking result. It is important to determine whether LOH is increased in these lesions, in order to test whether LOH is related to the phenotype.*

We thank the reviewer for this suggestion; however, we respectfully disagree that it is important to corroborate the anti-Fzd4 data with analysis of *Ptch* LOH. We show that the effects of perinatal anti-Fzd4 treatment on tumorigenesis phenocopies several key features associated with genetic Ndp/Fzd4 inactivation, including effects on tumor latency, incidence, lesion induction and vascularization (Figure 7). Including *Ptch* LOH analysis in this context will not change the main conclusion of the experiment, namely that genetic as well as acute inactivation of the Ndp/Fzd4 axis accelerates Shh MB, and will, therefore, not advance the study. Moreover, the experiment that the reviewer is requesting would require nearly a year to complete, which is far longer than the time allotted for submitting a revised manuscript. However, the reviewer’s comment raises the important issue of whether the tumor inhibitory effect of Ndp/Fzd4 signaling extends to models of Shh MB that do not require *Ptch* LOH for progression. To address this question we used the Smo-A1 mouse, a model where Shh MB is driven by expression of an oncogenic form of Smoothened in GNPS and is independent of *Ptch* LOH (Hallahan et al., 2004). We find that tumorigenesis in this model is also dramatically accelerated on an Ndp deficient background (revision, Figure 8). Based on this data we conclude that the tumor inhibitory effects of Ndp signaling extend to oncogene-driven Shh MB.

PLVAP was upregulated in endothelial cells when NDP-FZD4 signaling was suppressed. PLVAP is known to be essential for formation of vascular fenestration. Does PLVAP upregulation increase fenestration in the mutant tumor vasculature?

PLVAP expression has been shown to correlate with blood vessel fenestration in established MB tumors (Phoenix et al., 2016).